# Diagnostic potential for a serum miRNA neural network for detection of ovarian cancer

Kevin M Elias[1,2,3], Wojciech Fendler[2,4,5], Konrad Stawiski[4], Stephen J Fiascone[1,2], Allison F Vitonis[2,6,7], Ross S Berkowitz[1,2], Gyorgy Frendl[2,3,8], Panagiotis Konstantinopoulos[2,9], Christopher P Crum[2,10], Magdalena Kedzierska[11], Daniel W Cramer[2,6,7], Dipanjan Chowdhury[2,5]*

[1]Division of Gynecologic Oncology, Department of Obstetrics and Gynecology , Brigham and Women's Hospital, Dana-Farber Cancer Institute, Boston, United States; [2]Harvard Medical School, Boston, United States; [3]Surgical ICU Translational Research Center, Brigham and Women's Hospital, Boston, United States; [4]Department of Biostatistics and Translational Medicine, Medical University of Lodz, Lodz, Poland; [5]Department of Radiation Oncology, Dana-Farber Cancer Institute, Boston, United States; [6]Obstetrics and Gynecology Epidemiology Center, Department of Obstetrics and Gynecology , Brigham and Women's Hospital, Boston, United States; [7]Department of Epidemiology, Harvard School of Public Health, Boston, United States; [8]Department of Anesthesiology, Perioperative and Pain Medicine, Brigham and Women's Hospital, Boston, United States; [9]Department of Medical Oncology, Dana-Farber Cancer Institute, Boston, United States; [10]Division of Women's and Perinatal Pathology, Department of Pathology, Brigham and Women's Hospital, Boston, United States; [11]Department of Clinical Oncology, Medical University of Lodz, Lodz, Poland

*For correspondence: dipanjan_chowdhury@dfci.harvard.edu

Competing interests: The authors declare that no competing interests exist.

**Abstract** Recent studies posit a role for non-coding RNAs in epithelial ovarian cancer (EOC). Combining small RNA sequencing from 179 human serum samples with a neural network analysis produced a miRNA algorithm for diagnosis of EOC (AUC 0.90; 95% CI: 0.81–0.99). The model significantly outperformed CA125 and functioned well regardless of patient age, histology, or stage. Among 454 patients with various diagnoses, the miRNA neural network had 100% specificity for ovarian cancer. After using 325 samples to adapt the neural network to qPCR measurements, the model was validated using 51 independent clinical samples, with a positive predictive value of 91.3% (95% CI: 73.3–97.6%) and negative predictive value of 78.6% (95% CI: 64.2–88.2%). Finally, biologic relevance was tested using in situ hybridization on 30 pre-metastatic lesions, showing intratumoral concentration of relevant miRNAs. These data suggest circulating miRNAs have potential to develop a non-invasive diagnostic test for ovarian cancer.
DOI: https://doi.org/10.7554/eLife.28932.001

## Introduction

Invasive epithelial ovarian cancer (EOC) is the leading cause of death from gynecologic cancer among women in developed countries (*Siegel et al., 2016*). Most women with EOC present with advanced stage disease, where 5 year survival rates average 25–30%, highlighting the need for an effective screening strategy. Unfortunately, two large-scale randomized clinical trials involving ultrasound and CA125, including the Prostate, Lung, Colorectal, and Ovarian Cancer (PLCO) trial and the

**eLife digest** Ovarian cancer is a major cause of cancer death among women. A woman's survival often hinges on doctors detecting the tumor before it has spread beyond the ovary. Unfortunately, most women with ovarian cancer are not diagnosed until they have symptoms – such as pelvic pain, bloating, swelling of the abdomen or appetite loss. By then, the disease has usually spread and is difficult to treat. There is currently no reliable test to diagnose ovarian cancer before symptoms emerge. Some tests measure proteins in the blood or use ultrasound images to identify ovary tumors. These tests usually still identify the disease too late. Sometimes they produce "false positive" results, which may cause women without cancer to undergo unnecessary surgery.

Many ovarian cancers have defects in small pieces of genetic information called microRNAs. These microRNAs impact the tumor in multiple ways, and cells release microRNAs into the blood. Testing a seemingly healthy women's blood for the same pattern of altered microRNAs found in women with ovarian cancer might be one way to detect the disease earlier.

Now, Elias et al. have identified a pattern of seven microRNAs in the blood that appears to predict ovarian cancer. In the experiments, a computer program searched for microRNA patterns in women with ovarian cancer. The program sifted through the microRNAs in blood from women with and without ovarian cancer. Over time, the computer program "learned" to identify a pattern of microRNAs found only in women with ovarian cancer. It then created a formula for identifying ovarian cancer based on seven of the microRNAs.

Elias et al. then verified that the formula accurately detected ovarian cancer by testing it on blood samples from more women with and without cancer. They also found the seven microRNAs in tiny ovarian cancer tumors collected from women. This suggests the formula might be able to detect even the smallest tumors. More studies are needed to determine when this cancer-linked pattern first emerges and confirm that this ovarian cancer-detection formula works. If the test is validated, it might be used to screen women who are at high risk for ovarian cancer because of mutations in the *BRCA1* and *BRCA2* genes.

DOI: https://doi.org/10.7554/eLife.28932.002

United Kingdom Collaborative Trial of Ovarian Cancer Screening (UKCTOCS) trial did not demonstrate a meaningful impact on overall survival from EOC (*Zhu et al., 2011*; *Jacobs et al., 2016*). These and other non-experimental longitudinal studies reaffirm CA125 can detect advanced disease but with poorer sensitivity for early stage and non-serous cancers. In addition, CA125 has limited specificity, with the majority of abnormal CA125 values being the result of non-gynecologic malignancies or benign gynecologic conditions (*Moss et al., 2005*). The hope that adding more biomarkers to CA125 would improve screening was not realized in a re-analysis of the PLCO data as well as a recent longitudinal study from the European Prospective Investigation of Nutrition and Cancer (*Zhu et al., 2011*; *Terry et al., 2016*). In a separate strategy to improve EOC outcome, several panels (which have CA125 as part of them) have received FDA approval to be used in the differential diagnosis of EOC to encourage referral of EOC cases to centers with greater expertise in cancer surgery and chemotherapeutic treatment (*Karst and Drapkin, 2010*). However, these have not been effective for early diagnosis.

Among the alternatives to serum proteins for the diagnosis or early detection of EOC, circulating microRNAs (miRNAs) have shown great potential (*Nakamura et al., 2016*). miRNAs are short (18–24 nucleotide) non-coding RNAs that regulate gene expression through post-transcriptional modification of mRNA transcripts. miRNAs have several advantages over protein measures: (1) PCR amplifies detection of rare transcripts in blood; (2) all miRNAs use the same units of measure, easing incorporation into multiplexed panels; and (3) miRNAs play a critical role in ovarian cancer biology, whereas the function of CA125 is unknown (*Deb et al., 2017*; *Katz et al., 2015*). Moreover, non-invasive sampling of circulating miRNAs has a clear advantage over analytes obtained through biopsy (*Wang et al., 2016*).

Preliminary studies have suggested that circulating miRNAs profiles are altered in women with ovarian cancer (*Nakamura et al., 2016*; *Chung et al., 2013*; *Langhe et al., 2015*; *Resnick et al., 2009*; *Zuberi et al., 2015*; *Samuel and Carter, 2016*). In addition, miRNAs have prognostic

significance for EOC survival (*Merritt et al., 2008*; *Bagnoli et al., 2016*; *Cramer and Elias, 2016*). However, efforts to develop a diagnostic signature based on circulating miRNAs have been hampered by issues regarding the best statistical approach to develop a model, reproducibility of miRNA measurement across technology platforms (e.g. qPCR, next generation sequencing, microarray), and the biologic heterogeneity of EOC (*Nakamura et al., 2016*). In this study, our objective was to develop a serum-based miRNA model for the diagnosis of ovarian cancer that could address these concerns and demonstrate the biologic and clinical relevance of this diagnostic tool.

## Results

To produce our diagnostic circulating miRNA signature from human sera, we constructed a study population of pre-treatment (prior to either surgery or chemotherapy) subjects comprising 179

**Table 1.** Demographics of patients in the model study populations.

| | ERASMOS (n = 60) | PMP/NECC (n = 119*) | p-value |
|---|---|---|---|
| Age, years, median (SD)[†] | 57 (9.8) | 56 (7.1) | 0.44 |
| CA-125, units/ml, median (SD) [†] | 155 (689.8) | 88.1 (1335.5) | 0.72 |
| Histology, n (%)[‡] | | | |
| Control | 0 (0) | 15 (12.6) | <0.0001 |
| Serous cystadenoma/cystadenofibroma | 7 (11.7) | 14 (11.8) | |
| Endometrioma | 0 (0) | 15 (12.6) | |
| Other benign lesion | 9 (15.0) | 0 (0) | |
| Borderline mucinous tumor | 2 (3.3) | 0 (0) | |
| Borderline serous tumor | 5 (8.3) | 15 (12.6) | |
| Stage I/II serous adenocarcinoma | 5 (8.3) | 20 (16.8) | |
| Stage III/IV serous adenocarcinoma | 19 (31.2) | 10 (8.4) | |
| Stage I/II clear cell/endometrioid adenocarcinoma | 6 (10.0) | 20 (16.8) | |
| Stage III/IV clear cell/endometrioid adenocarcinoma | 0 (0) | 10 (8.4) | |
| Mucinous adenocarcinoma | 1 (1.7) | 0 (0) | |
| Other ovarian cancer | 10 (10.0) | 0 (0) | |
| Stage, n (%)[‡] | | | |
| Not applicable | 16 (26.7) | 59 (49.6) | <0.0001 |
| I | 9 (15.0) | 22 (18.5) | |
| II | 8 (13.3) | 18 (15.1) | |
| III | 19 (31.2) | 18 (15.1) | |
| IV | 8 (13.3) | 2 (1.7) | |
| Grade, n (%)[‡] | | | |
| Not applicable | 16 (26.7) | 44 (37.0) | 0.07 |
| Borderline | 7 (11.7) | 15 (12.6) | |
| 1 (well-differentiated) | 6 (10.0) | 12 (10.1) | |
| 2 (moderately differentiated) | 3 (5.0) | 12 (10.1) | |
| 3 (poorly differentiated) | 28 (46.7) | 36 (30.3) | |

ERASMOS – Effects of Regional Analgesia on Serum miRNA after Oncology Surgery Study

PMP – Pelvic Mass Protocol

NECC – New England Case Control study

*15samples from NECC, 114 samples from PMP

[†]student's t-test

[‡]chi-square test

DOI: https://doi.org/10.7554/eLife.28932.004

women selected from three independent prospective studies (ERASMOS, PMP, and NECC) (*Table 1*). ERASMOS contributed consecutive cases presenting for evaluation of an adnexal mass, while PMP allowed enrichment of the population for specific histopathologic diagnoses. NECC added healthy controls age-matched to PMP. After completing small RNA sequencing on the sera, subjects were randomly assigned into model training and testing sets (*Figure 1*). After the randomization, the training and testing sets were demographically similar, and there were no differences in the distribution of histopathological diagnoses between the sets (*Table 2*).

We then deployed a series of statistical tools, including machine-learning approaches to analyze the miRNA-seq data to create an algorithm with the best performance for discriminating cases of ovarian cancer from either benign tumors, non-invasive ('borderline') tumors, or healthy controls. This began by using three different potential strategies for selecting miRNA variable inputs to build the models: significance-based (by t-test), correlation-based feature subset, or expression fold change (*Table 3*). Each miRNA variable list method was entered into one of 11 different models, which were compared both by AUC (*Table 4*) as well as sensitivity and specificity (*Figure 2*).

Although many of the models performed well, the neural network model employing miRNA expression fold changes was the only model to meet our pre-specified statistical objective with an AUC of 0.90 (95% CI: 0.81–0.99; p=0.03 over a theoretical AUC of 0.75). The network consisted of 14 individual miRNAs with seven neurons in the hidden layer (*Source code 1*). As the network relied on complex interactions between miRNA levels we tested whether its performance was not biased by batch adjustment performed at the initial step of the analysis. The neural network worked equally well on the adjusted and unadjusted raw datasets with an AUC of 0.93 (95%CI: 0.89–0.98) on the

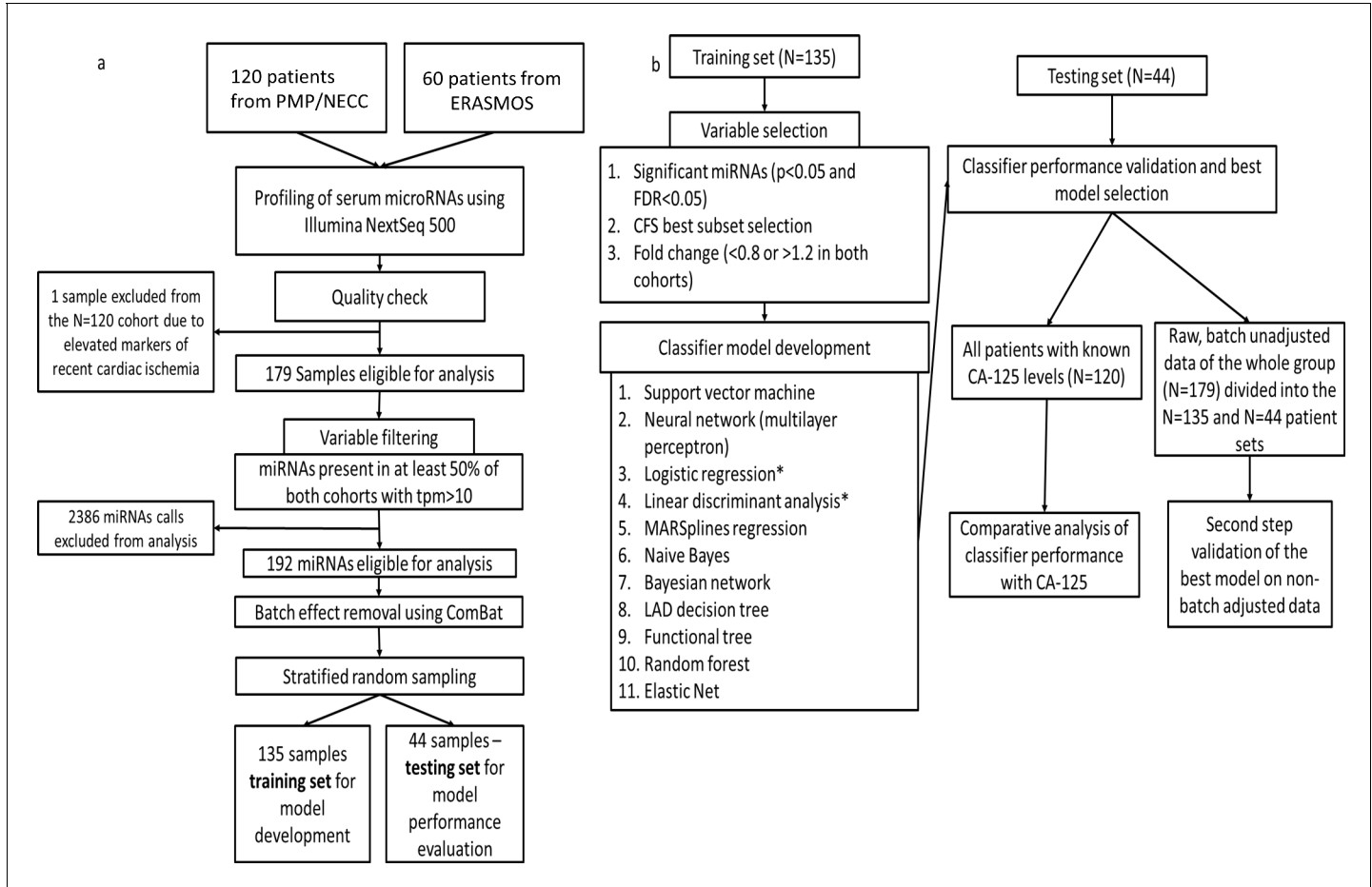

**Figure 1.** Flowchart of study design. (a) Protocol for miRNA sequencing, filtering, batch adjustment and separation into the training and testing sets. (b) Protocol for model development and testing.
DOI: https://doi.org/10.7554/eLife.28932.003

**Table 2.** Demographics of patients after stratified random sampling into training and testing sets.

| | Training (n = 135) | Testing (n = 44) | p-value |
|---|---|---|---|
| **Age, years, median (SD) *** | **56 (8.1)** | **56 (8.3)** | **1.0** |
| CA-125, units/ml, median (SD) * | 126.5 (1193.5) | 105.6 (577.8) | 0.91 |
| Pathology, n (%)† | | | 1.0 |
| Control | 11 (8.1) | 4 (9.1) | |
| Benign lesions | 34 (25.2) | 11 (25.0) | |
| Borderline tumors | 16 (11.9) | 5 (11.4) | |
| Stage I/II invasive cancers | 41 (30.4) | 12 (27.3) | |
| Stage III/IV invasive cancers | 33 (24.4) | 12 (27.3) | |

*student's t-test
†chi-square test
DOI: https://doi.org/10.7554/eLife.28932.005

training and 0.90 (95%CI 0.80–0.99) on the testing set (*Figure 3*; *Supplementary file 1A* [by model] and 1B [by sample]). In post-hoc secondary analyses, the neural network worked equally well for older and younger patients, serous and non-serous histologies, and early and advanced stage disease (*Supplementary file 2A-C*).

Serum CA125 data were available for 120 subjects (*Supplementary file 1B* and *3A*). Among these, the neural network (AUC 0.93; 95% CI 0.88–0.97) significantly outperformed CA125 (AUC 0.74; 95% CI 0.65–0.83; p=0.001; *Figure 4*). The primary advantage of the neural network over CA125 was avoiding false positives (8/43 for the neural network versus 23/43 for CA125; p=0.002) (*Supplementary file 2A*). Notably, the neural network and CA125 levels were independent of one another (*Figure 4—figure supplement 1*; *Supplementary file 3B*). We tested using the neural network and CA125 in a tiered testing strategy, subjecting all negative neural network algorithm results to a second review with CA125, but found this would increase the probability of a false positive test result from 4.2% (5/120) to 19.2% (23/120) and a false negative rate from 5.8% (7/120) to 13.3% (16/120) (*Figure 4—figure supplement 2*). The alternative of initial screening with CA125 followed by neural network yielded only three additional correctly diagnosed cases of invasive cancer at the expense of 19 additional false positive results.

**Table 3.** miRNA variables used in model building identified through univariate testing

| Significance-based selection | Correlation-based feature subset selection | Expression fold change selection |
|---|---|---|
| miR-29a-3p | miR-16-2-3p | miR-23b-3p |
| miR-30d-5p | miR-200a-3p | miR-29a-3p |
| miR-200a-3p | miR-200c-3p | miR-32–5 p |
| miR-200c-3p | miR-320b | miR-92a-3p |
| miR-320d | miR-320d | miR-150–5 p |
| miR-320c | | miR-200a-3p |
| miR-450b-5p | | miR-200c-3p |
| miR-203a | | miR-203a |
| miR-486–3 p | | miR-320c |
| miR-1246 | | miR-320d |
| miR-1307–5 p | | miR-335–5 p |
| | | miR-450b-5p |
| | | miR-1246 |
| | | miR-1307–5 p |

DOI: https://doi.org/10.7554/eLife.28932.007

**Table 4.** Performance of the eleven statistical models on the testing set by variable selection method. Results are shown for the testing set.

| Statistical model | Variable selection method | | |
| --- | --- | --- | --- |
| | Significance-based variable subset AUC (95% CI) | Correlation-based feature selection subset AUC (95% CI) | Fold change-based variable subset AUC (95% CI) |
| Linear discriminant analysis | 0.80 (0.66–0.93) | 0.76 (0.62–0.90) | 0.78 (0.64–0.92) |
| Logistic regression | 0.81 (0.68–0.94) | 0.75 (0.61–0.90) | 0.82 (0.70–0.94) |
| Neural network | 0.84 (0.72–0.96) | 0.75 (0.60–0.89) | 0.90 (0.81–0.99) |
| Support vector machine | 0.77 (0.63–0.91) | 0.73 (0.58–0.87) | 0.77 (0.63–0.91) |
| Multivariate adaptive regression splines | 0.57 (0.40–0.74) | 0.66 (0.49–0.82) | 0.73 (0.58–0.88) |
| Naive Bayes classifier | 0.75 (0.60–0.89) | 0.68 (0.52–0.84) | 0.75 (0.60–0.89) |
| Least Absolute Deviation regression tree | 0.77 (0.63–0.91) | 0.61 (0.44–0.78) | 0.69 (0.53–0.84) |
| Functional tree | 0.78 (0.64–0.91) | 0.77 (0.63–0.91) | 0.68 (0.52–0.84) |
| Bayesian network | 0.72 (0.56–0.87) | 0.67 (0.52–0.83) | 0.72 (0.56–0.87) |
| Random forest | 0.78 (0.64–0.91) | 0.71 (0.56–0.86) | 0.76 (0.62–0.90) |
| Elastic net | 0.80 (0.67–0.93) | 0.76 (0.62–0.90) | 0.79 (0.66–0.92) |

DOI: https://doi.org/10.7554/eLife.28932.008

The specificity of the neural network algorithm for the diagnosis of ovarian cancer was tested using an external, independent, dataset previously published by Keller, *et al* (*Keller et al., 2011*). These data were generated via a third technology platform, probe-based microarray, which fortunately contained all 14 miRNAs from our original signature, allowing for 1:1 mapping without exclusions (*Supplementary file 4A* and *Supplementary file 6*). The neural network perfectly classified

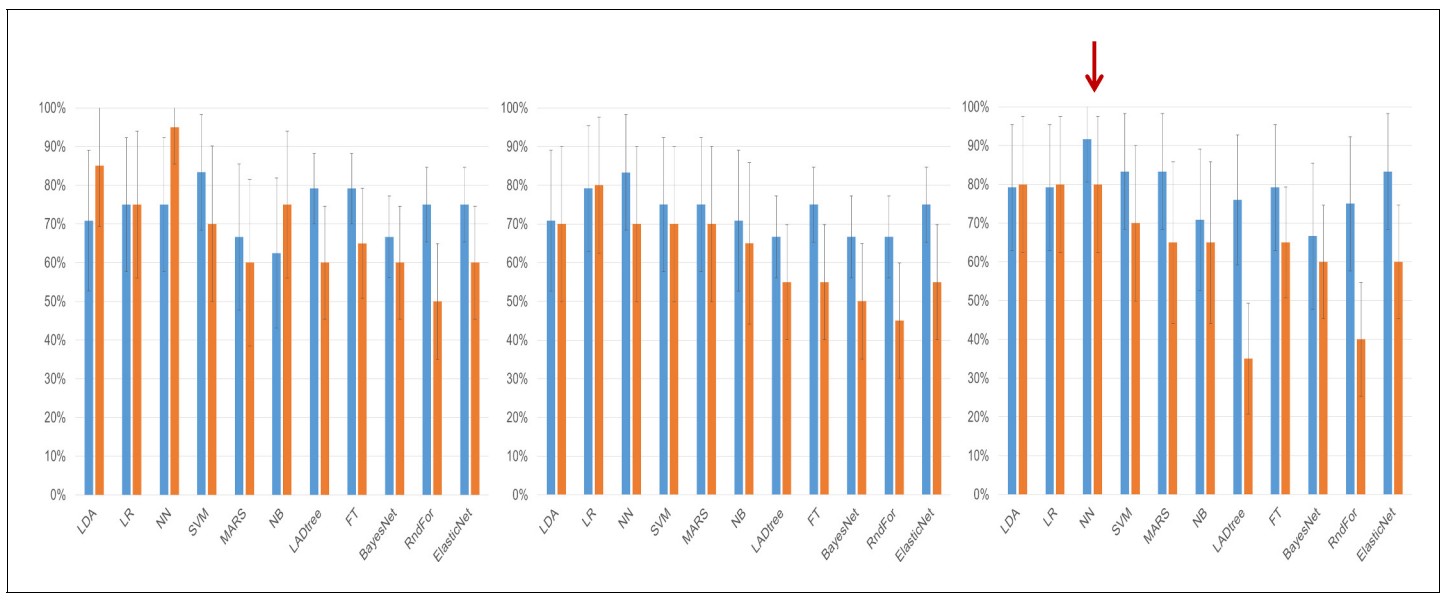

**Figure 2.** Clinical performance characteristics of the tested models. Sensitivity (blue bars) and specificity (orange bars) of the classifiers on the testing set depending on the method of variable selection. Whiskers denote 95% Confidence Intervals. (**a**) – Performance of models created on the subset of miRNAs selected using the significance-based filter. (**b**) Performance of models created on variables selected using the CFS subset algorithm. (**c**) Performance of models created using variables selected by the fold change-based filter. The red arrow denotes the model with the best performance characteristics, the neural network analysis using the fold change-based filter variable.

DOI: https://doi.org/10.7554/eLife.28932.006

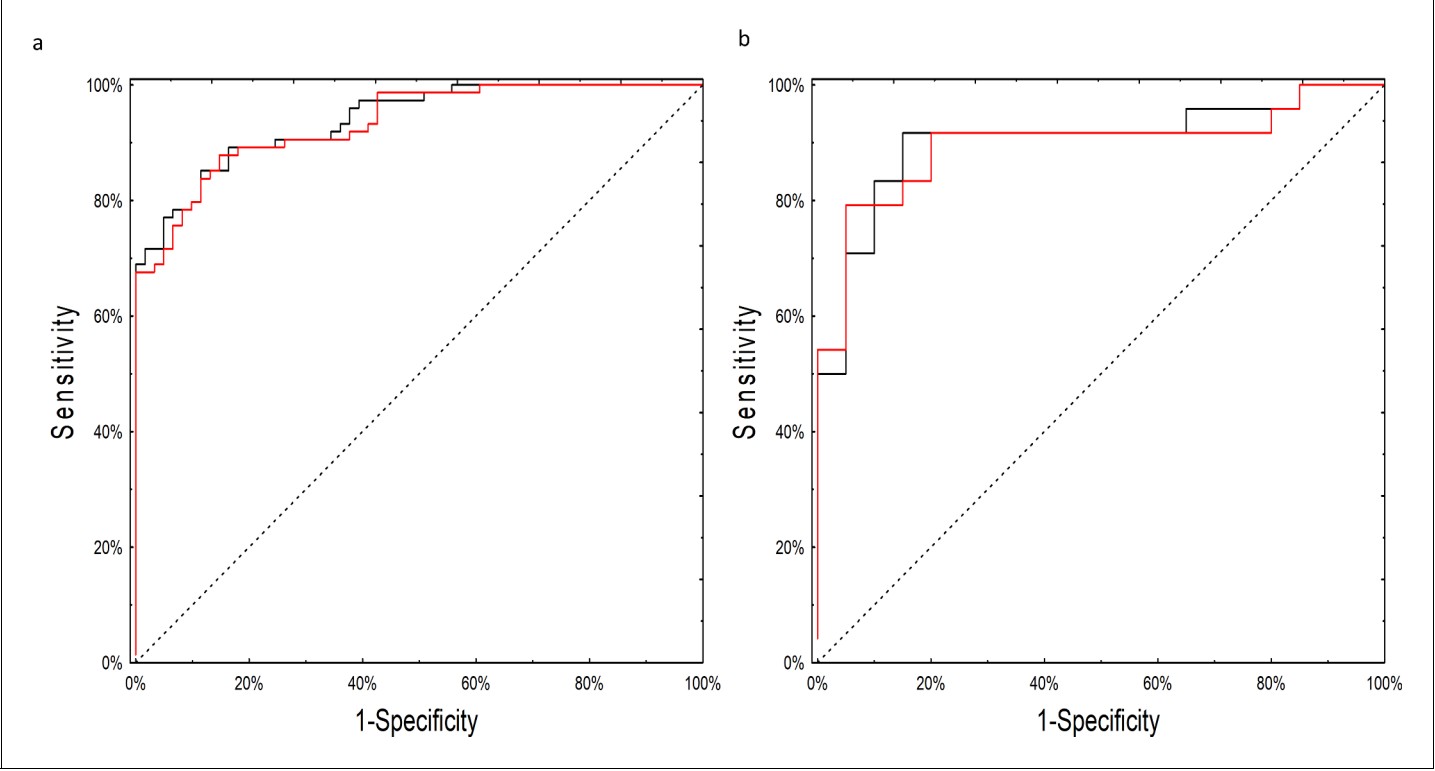

**Figure 3.** ROC curves for the neural network analysis. (**a**) Performance of the neural network on the training set of raw, non-batch-adjusted data (red line) and in the batch-adjusted training set (black line) (**b**) Performance of the neural network on raw (red line) and batch-adjusted (black line) data in the testing set.

DOI: https://doi.org/10.7554/eLife.28932.009

patients in the training set (AUC 1.00, 95% CI 1.00–1.00) and provided very good discriminatory power on the testing set (AUC 0.93, 95% CI 0.81–1.00), with an overall sensitivity of 75% and specificity of 100%. The signature was specific to ovarian cancer compared to all other diagnoses, as it did not show any clinically-efficient diagnostic capabilities for any of the 12 other morbidities analysed in the set and showed good performance in distinguishing ovarian cancer samples against all other diagnoses combined (AUC 0.92, 95% CI 0.82–1.00) (*Figure 5*).

Having established our miRNAs of interest using next generation sequencing, we next sought to validate the sequencing data across technology platforms by measuring the miRNAs from the neural network using qPCR. While small RNA sequencing is a more robust technology for miRNA discovery, qPCR is a more time efficient and cost-effective diagnostic tool. For this we used 120 samples from PMP and NECC for which we had excess RNA. We internally validated the 14 miRNAs in the neural network (plus an additional nine potential reference miRNAs derived from the sequencing data) by qPCR and recalibrated the algorithm to accept qPCR inputs (*Supplementary file 6*). We then performed a global sensitivity analysis on the best neural network for qPCR data and iteratively removed the variables which did the least in terms of improving the classifier's performance. This reduced the neural network to only seven miRNAs (miR-29a-3p, miR-92a-3p, miR-200c-3p, miR-320c, miR-335–5 p, miR-450b-5p, and miR-1307–5 p) plus four normalizers (miR-423–3 p, miR-191–5 p, miR-221–3 p, and miR-103a-3p). To increase the statistical power of this qPCR-based classifier and create a fully locked-down model for clinical application, we added 205 more samples from PMP and NECC, including more than 100 additional healthy controls, to create a 325 subject population for qPCR model development (*Table 5*).

These samples were randomized 3:1 into training and testing sets to create a neural network. The resulting network performed well with an AUC 0.89 on the training set and AUC 0.80 on the testing set.

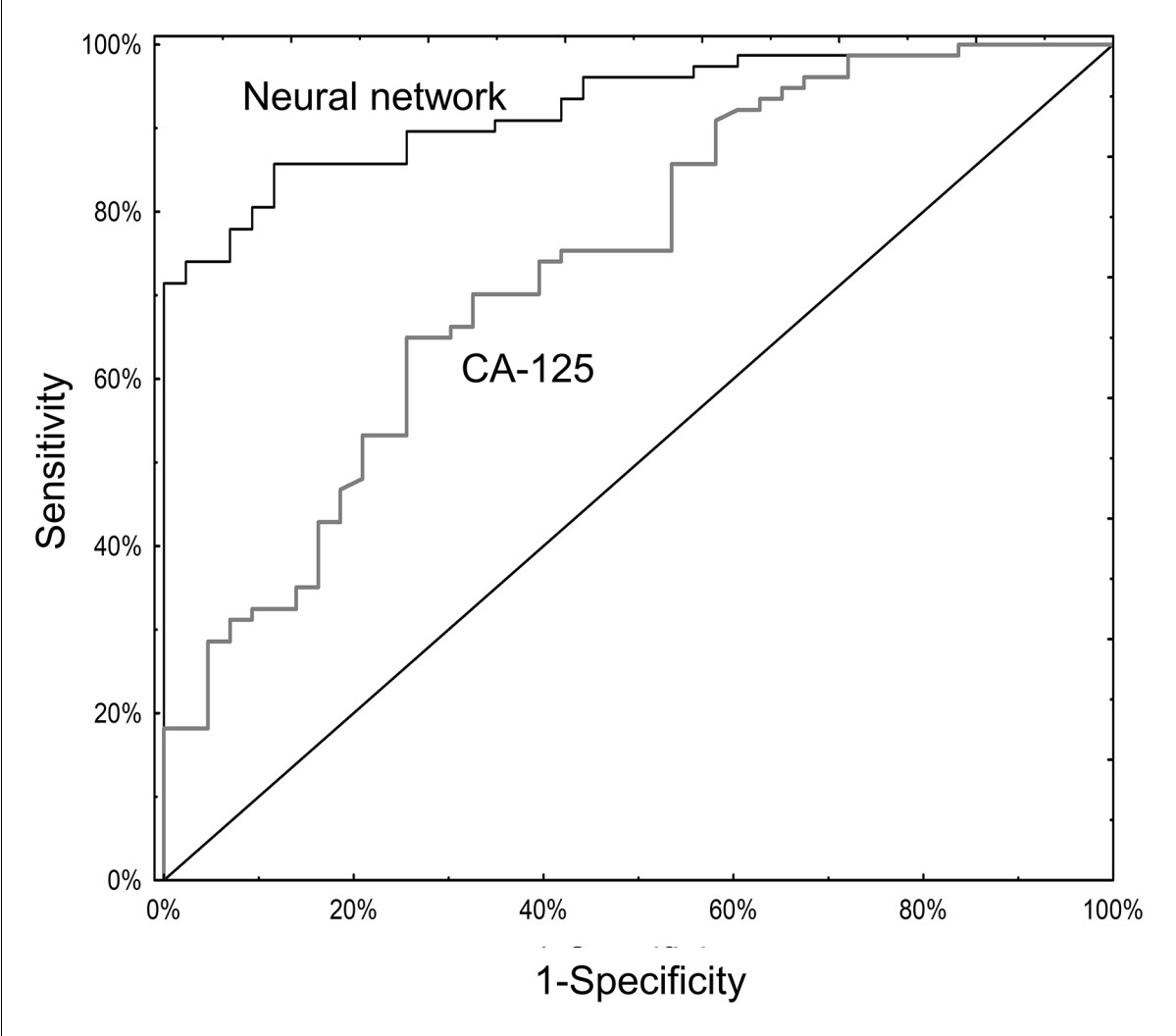

**Figure 4.** ROC curves for neural network analysis compared to CA-125. The neural network (AUC 0.93; 95% CI 0.88–0.97) significantly outperformed CA125 (AUC 0.74; 95% CI 0.65–0.83) in terms of overall operating characteristics (p=0.001).

DOI: https://doi.org/10.7554/eLife.28932.010

The following figure supplements are available for figure 4:

**Figure supplement 1.** Correlations between the miRNAs (vertical axes) of the neural network and CA-125 (horizontal axes) in the cancer (red markers) and benign/borderline/control (blue markers) groups.

DOI: https://doi.org/10.7554/eLife.28932.011

**Figure supplement 2.** Performance of a two-tiered algorithm for ovarian cancer diagnosis incorporating both the neural network (NN) and a CA-125 cut-off of 35 U/ml.

DOI: https://doi.org/10.7554/eLife.28932.012

We then tested the clinical performance of the final, locked-down diagnostic test on a completely independent external sample set collected from 51 preoperative patients treated in Lodz, Poland (*Table 6*). In this population, the neural network had a positive predictive value of 91.3% (95% CI: 73.3–97.6%) and a negative predictive value of 78.6% (95% CI: 64.2–88.2%) with an AUC of 0.85 (*Figure 6*).

Ideally, a serum biomarker should have biologic relevance to the clinical disease. To this end, we returned to the ERASMOS patient set to examine if the expression levels of the miRNAs changed in the cancer patients after surgical cytoreduction. Among the patients with ovarian cancer in the study, 27 had both preoperative and postoperative serum miRNAs profiled. These included 4/7 target miRNAs in the qPCR neural network model. Circulating levels of all three miRNAs decreased

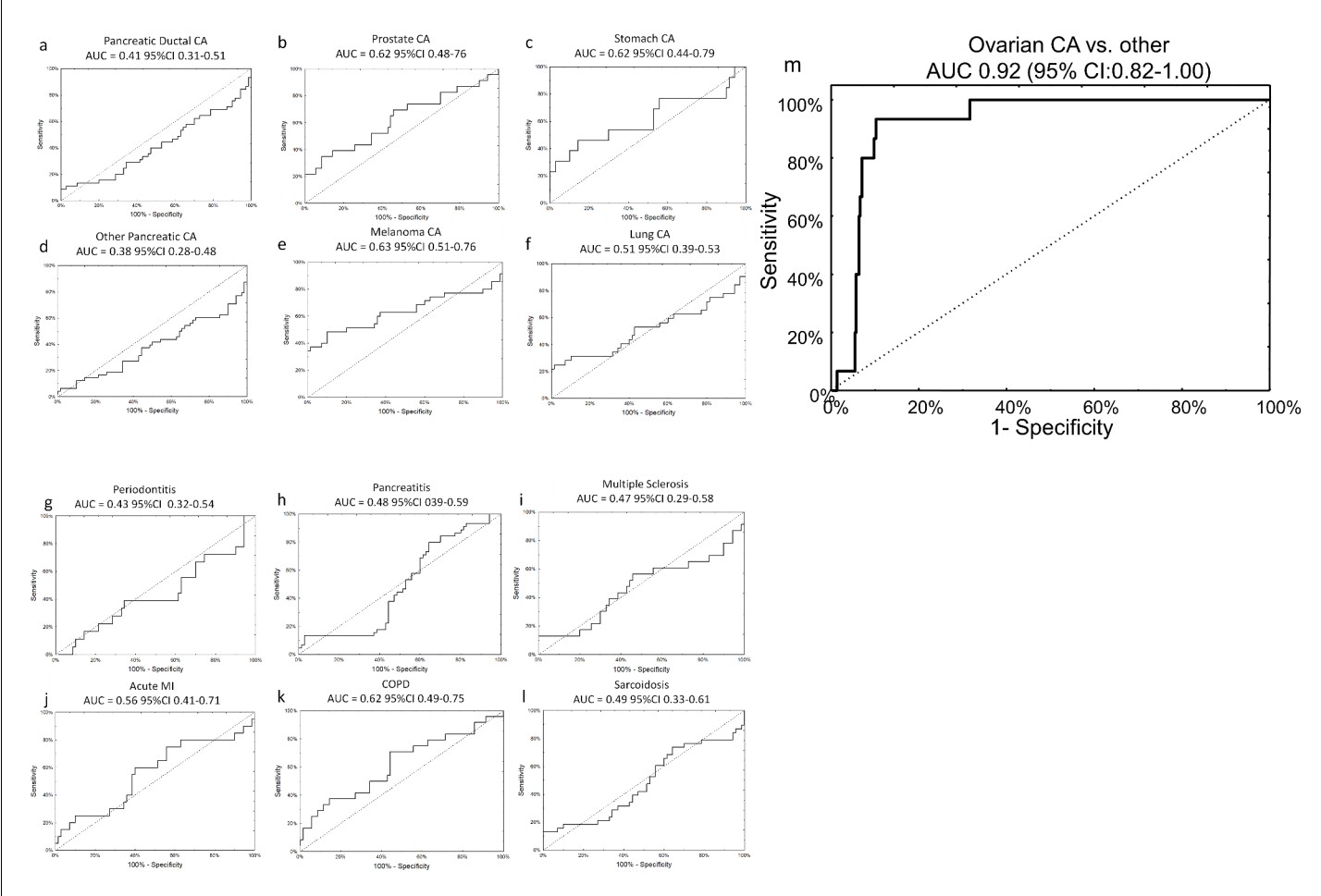

**Figure 5.** Specificity of miRNA signature for ovarian cancer compared to other diagnoses. The neural network 14 miRNA signature did not separate any other diagnoses from the control group in the published dataset by Keller, *et al* [13]. The study also included 70 healthy controls. The number of subjects (**n**) denotes the number of cases of the given diagnosis in the Keller, *et al* dataset. (**a**) Pancreatic ductal cancer (n = 45); (**b**) Prostate cancer (n = 23); (**c**) Stomach cancer (n = 13); (**d**) Other pancreatic cancers (n = 48); (**e**) Melanoma (n = 35); (**f**) Lung cancer (n = 32); (**g**) Periodontitis (n = 18); (**h**) Pancreatitis (n = 38); (**i**) Multiple sclerosis (n = 23); (**j**) Acute MI (n = 20); (**k**) Chronic obstructive pulmonary disease (n = 24); (**l**) Sarcoidosis (n = 45). (**m**) Overall, neural network was highly specific for ovarian cancer cases against all other diagnoses (i.e. healthy controls or other cancers).
DOI: https://doi.org/10.7554/eLife.28932.013

within 72 hr of tumor removal, with significant changes for miR-200a-3p and miR-200c-3p (*Figure 7A–D*).

We also wanted to test if the miRNAs were in fact coming from the earliest lesions of this disease. For this, we assembled paraffin-embedded tissue sections from independent sets of 15 cases of serous tubal intraepithelial carcinomas and 15 Stage I high grade (serous or Grade three endometrioid) epithelial ovarian cancers. Immunohistochemistry was performed on sequential sections for TP53 and Ki67 to highlight the lesions. We then performed in situ hybridization for three of the miRNAs in our neural network; mir-200c-3p, mir-335–5 p, and mir-92a-3p (*Figure 8*). In 100% of the samples, there was complete overlap between lesional cells and the miRNAs crucial for neural network performance, suggesting that the miRNAs detected in the serum are present even in early lesions in the fallopian tube epithelium and raising the possibility of detection of pre-metastatic disease.

Finally, we have constructed a web calculator (http://biostat.umed.pl/ovaries) to demonstrate how to use these models. The calculator accepts various inputs describing on the method of circulating miRNA quantification (sequencing, qPCR, or microarray) and returns the estimated probability of ovarian cancer for a given patient.

**Table 5.** Clinical characteristics of the qPCR model set.

| Characteristic | qPCR model set (N = 325) |
|---|---|
| Age, years, median (SD) | 58.0 (10.1) |
| Grade, n (%) | |
| Borderline | 15 (4.6) |
| 1 | 21 (6.4) |
| 2 | 27 (8.3) |
| 3 | 100 (30.8) |
| unspecified | 10 (3.1) |
| Not applicable | 150 (46.2) |
| FIGO Stage, n (%) | |
| I/II | 75 (23.1) |
| III/IV | 83 (25.5) |
| Not applicable | 167 (51.4) |
| Histology, n (%) | |
| Control | 123 (37.8) |
| Serous cystadenoma/cystadenofibroma | 14 (4.3) |
| Endometrioma | 15 (4.6) |
| Borderline serous tumor | 15 (4.6) |
| Serous adenocarcinoma | 100 (30.8) |
| Endometrioid/clear cell adenocarcinoma | 48 (14.8) |
| Mucinous adenocarcinoma | 10 (3.8) |

DOI: https://doi.org/10.7554/eLife.28932.014

## Discussion

We have described the development of a diagnostic model for ovarian cancer using sequencing of circulating miRNA. This is the first study in ovarian cancer to combine next generation sequencing technology for serum miRNA with machine learning techniques. Not only does sequencing provide greater sensitivity for miRNA detection than other methods, but expression levels of various miRNAs are not linearly related and relationships among miRNAs tend to be obscured by more basic statistical approaches. The neural network as presented has several advantages over a traditional biomarker like CA125. The neural network recognized more Stage I/II ovarian cancers and had significantly fewer false positives. This likely reflects an ability to discriminate relevant biology more than to quantify tumor burden. For example, the neural network correctly classified 35/43 (81%) borderline tumors as being non-invasive neoplasms, compared to just 20/43 (47%; p=0.002) for CA125. An additional strength of our study is the incorporation of multiple independent datasets. The ERAS-MOS specimens were obtained from cases enrolled sequentially, reflecting the natural frequency of different ovarian tumor subtypes in the clinical population, including the fact that most women with invasive ovarian cancer present with advanced stage disease. The Pelvic Mass Protocol samples allowed us to enrich the study population for less common clinical cases that would be expected to confound a conventional screening algorithm, including benign complex ovarian masses, borderline tumors, early stage cancers, and non-serous histologic subtypes. NECC provided age-matched healthy controls. The specificity of our model was tested using a publicly available dataset from Keller, *et al* where we showed that the neural network performed well across disease stages, histologic subtypes, and diagnostic platforms. This ability to specifically identify ovarian cancers and discriminate ovarian cancer from other diagnoses sets the current work apart from prior miRNA studies (*Nakamura et al., 2016*; *Chung et al., 2013*; *Resnick et al., 2009*; *Zuberi et al., 2015*; *Häusler et al., 2010*; *Zheng et al., 2013*). Finally, we tested our signature using a completely

**Table 6.** Clinical characteristics of the external validation set.

| Characteristic | Polish external validation set (N = 51) |
|---|---|
| **Age, years, median (SD)** | **55.5 (16.1)** |
| Grade, n (%) | |
| Borderline | 4 (7.8) |
| 1 | 2 (3.9) |
| 2 | 7 (13.7) |
| 3 | 13 (25.5) |
| unspecified | 3 (5.9) |
| Benign | 22 (43.1) |
| FIGO Stage, n (%) | |
| I | 7 (13.7) |
| II | 3 (5.9) |
| III | 18 (35.3) |
| IV | 1 (2.0) |
| Benign | 22 (43.1) |
| Histology, n (%) | |
| Serous cystadenoma/cystadenofibroma | 6 (11.8) |
| Endometrioma/endometriosis | 10 (19.6) |
| Mature teratoma | 6 (11.8) |
| Borderline serous tumor | 2 (3.9) |
| Borderline seromucinous tumor | 2 (3.9) |
| Serous adenocarcinoma | 4 (7.8) |
| Mucinous adenocarcinoma | 1 (2.0) |
| Endometrioid adenocarcinoma | 1 (2.0) |
| Clear Cell Adenocarcinoma | 9 (17.6) |
| Mixed adenocarcinoma | 3 (5.9) |
| Adenocarcinoma unspecified | 7 (13.7) |

DOI: https://doi.org/10.7554/eLife.28932.016

external, independent set of samples from Poland, showing that in a clinical sample set the test performed well without additional modifications.

There appears to be biologic relevance to the serum miRNAs in the neural network. The rapid change in circulating levels after surgical cytoreduction for mir-200a and mir-200c suggests these are being produced actively by tumors. Although other miRNAs did not have as great of a decrease, this may reflect differing half-lives for different miRNA species. In future work, it would be interesting to measure changes over a longer time frame than 72 hr, but that was the endpoint for ERASMOS, which is an anesthesia-focused study. We also demonstrated expression of several miRNAs from the neural network in pre-metastatic lesions. This both confirms prior work suggesting that these miRNAs are detectable in advanced ovarian cancers specimens and adds the new finding that these miRNAs are expressed in very early stage and even pre-invasive lesions (*Bagnoli et al., 2016*). Future work will examine the kinetics of these miRNA changes in tumor pathogenesis.

The phase II specimens used in this study are like those used to support development of assay panels subsequently approved for the differential diagnosis of ovarian cancer vs. a benign pelvic mass. The first panel, named OVA1, was approved by the FDA in 2009 and consisted of 5 analytes including CA125 (*Zhang et al., 2004*; *Ueland et al., 2011*). While those authors emphasized the assay's negative predictive value of 95% (when combined with physician assessment), the assay had an AUC of only 0.80 (95% CI: 0.73–0.88) for pre-menopausal women and 0.82 (95% CI: 0.77–0.87) for post-menopausal women. The second panel was approved in 2011 and consisted of just two

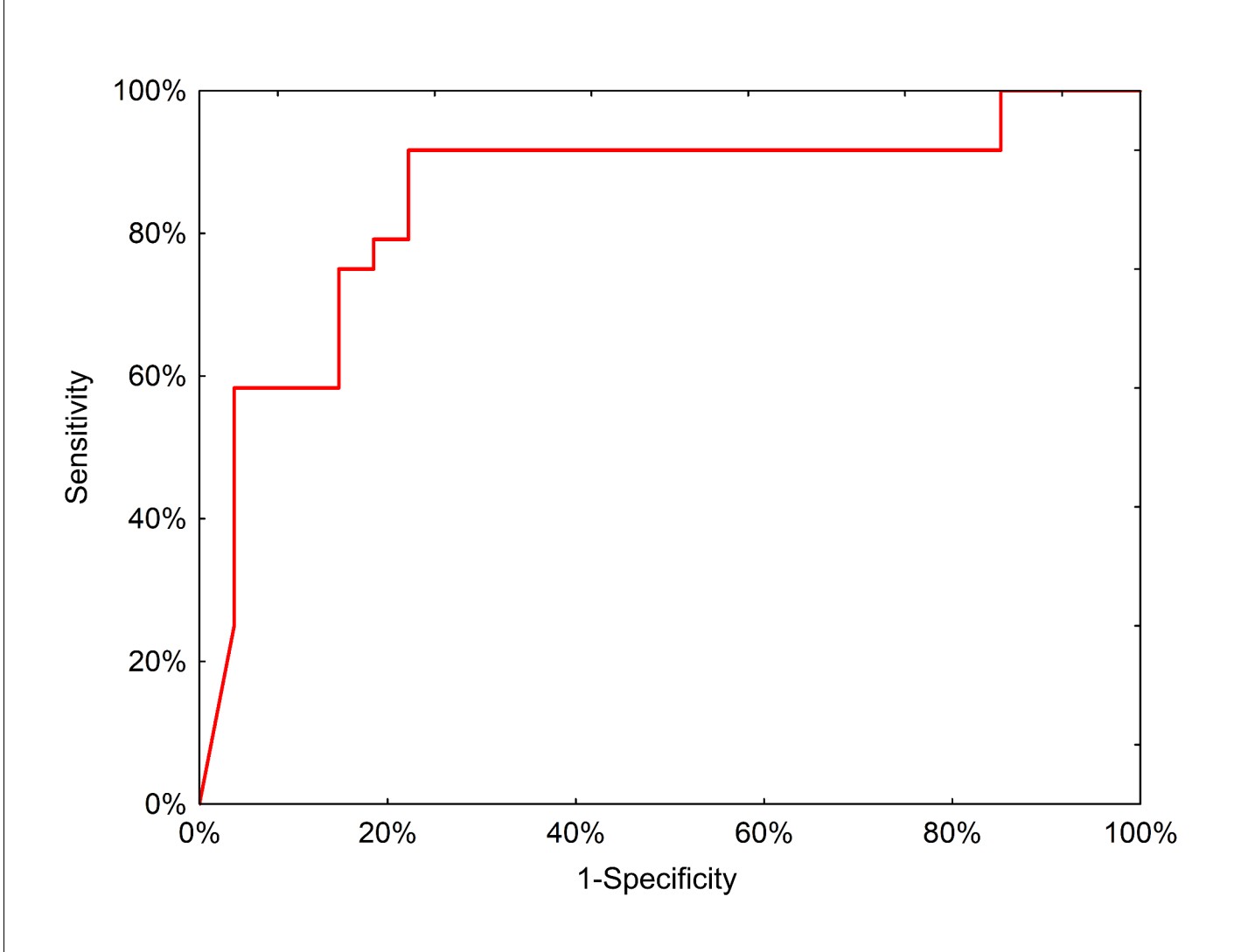

**Figure 6.** ROC curve for neural network analysis using qPCR inputs from the clinical test set.
DOI: https://doi.org/10.7554/eLife.28932.015

markers, CA125 and HE4, combined with menopausal status (*Moore et al., 2010*). While the ROMA algorithm had an overall AUC for discriminating cancer from benign tumors of 0.91 (95% CI: 0.88–0.94), this was in the setting of including borderline tumors as malignancies. Moreover, the positive predictive value of the test for distinguishing benign masses from Stage I/II EOC was only 0.27. In 2016, the FDA approved an updated version of the OVA1 test which retained CA125 but replaced 2 of the markers with HE4 and FSH (*Coleman et al., 2016*). This improved the overall AUC to 0.92 (95% CI: 0.89–0.96) for the assay alone and 0.94 (95% CI: 0.91–0.97) when combined with physician assessment, although 80% of the tumors in this study were benign. Although the above panels included some clinical information and therefore are not equivalent to our panel, we point out that the AUC of our panel to distinguish a malignant from benign pelvic mass was similar, while not including borderline tumors as positive results and agnostic to clinical or imaging information. As timely referral to a gynecologic oncologist is a strong predictor of ovarian cancer survival, we believe that there is a role for a test based on blood markers alone (*Earle et al., 2006*).

FDA approval of the various panels for use in the differential diagnosis of pelvic masses did not extend to their use in the general population. Based upon the results of the PLCO and UKCTOCS randomized clinical trials (or so called 'phase 4') the US Preventive Services Task Force and the Society of Gynecologic Oncology (SGO) have not recommended routine screening for ovarian cancer

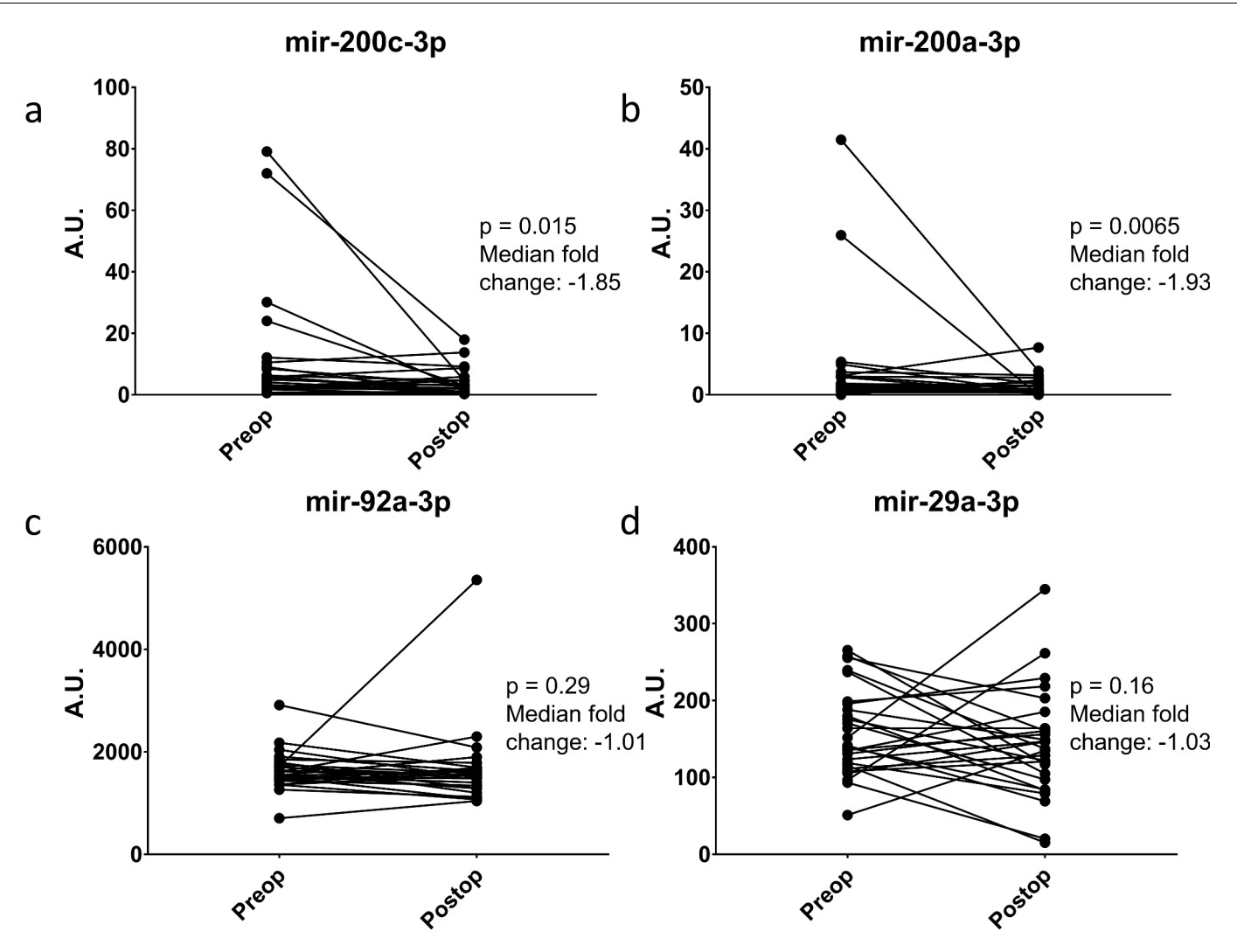

**Figure 7.** Change in miRNA expression from preop to post-operative day three after surgical cytoreduction. n = 27.
DOI: https://doi.org/10.7554/eLife.28932.017

(*Zhu et al., 2011*; *Skates et al., 2001*). However, screening with CA125 and transvaginal ultrasound is recommended by the National Comprehensive Cancer Network guidelines and the SGO for women with known hereditary syndromes of ovarian cancer (such as women with germline *BRCA1/2* mutations), even though there is currently no evidence that this screening strategy improves survival in elevated risk populations (*Schorge et al., 2010*).

Recent studies (*Chung et al., 2013*; *Langhe et al., 2015*; *Resnick et al., 2009*; *Zuberi et al., 2015*) have identified circulating (serum/plasma) miRNAs that are altered in ovarian carcinomas, and there is limited overlap with miRNAs that emerged from our analysis. One possible cause of this difference is the limited number of samples examined in these studies. For example, in Langhe et al, a training set of 5 serous ovarian carcinomas and five benign serous cystadenomas were selected for the initial experiments. The validation set was 20 serous ovarian carcinomas and 20 benign serous cystadenomas. In Resnick et al, 28 ovarian carcinoma patients and 15 healthy controls were used to identify to differential expression of circulating miRNAs. Such limited numbers diminish the statistical robustness of the results. Another possible cause for the differences is the miRNA expression profiling platform. Recently a study (*Mestdagh et al., 2014*) systematically compared 12 different miRNA expression platforms. Specifically, for serum miRNAs there was a 12-fold difference between the highest and lowest number of detected miRNA when identical samples were profiled by different

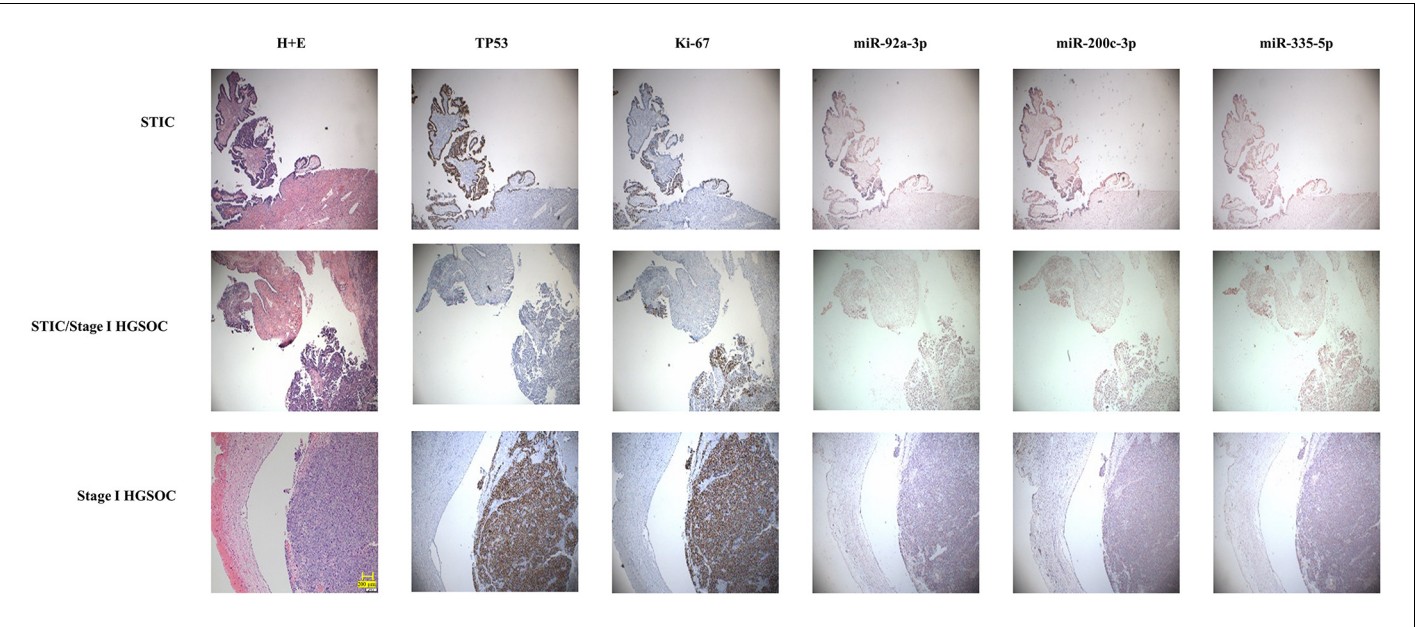

**Figure 8.** In situ expression of selected miRNAs from the serum signature. Sections of fallopian tubes showing serous tubal intraepithelial carcinoma (STIC) lesions and Stage I high grade serous ovariancancer (HGSOC). Lesional cells are indicated by TP53 and Ki-67 staining. (top) STIC lesion in continuity with normal fallopian tube. 20x. (middle) STIC lesion in continuity with normal fallopian tube and invasive cancer with p53-null lesion. 10x. (bottom) HGSOC intraluminal to the fallopian tube. 10x.

DOI: https://doi.org/10.7554/eLife.28932.018

platforms. According to this report the LNA-based platform from Exiqon has the highest specificity but maybe limited for sensitivity thereby for detection. To circumvent both these concerns we started with next-generation sequencing of 179 samples which captures all small RNAs and addresses any issues of detection or specificity due to limitations of platforms. Next, we did validation using the Exiqon qRT-PCR platform on 325 local samples, and a further validation using an additional cohort of 51 samples from Poland. The large number of samples along with the methodologies used for identification and validation of the circulating miRNAs in our study provides strong support for our conclusions and distinguishes our work from prior reports.

Our study does have several limitations. Whether our miRNA panel will prove useful in the differential diagnosis of early detection will require further study in the following areas. First, additional study is necessary to determine whether integrating clinical risk factors could further improve its performance. Second, confirmation in other phase II data sets are necessary to validate our study results and demonstrate its generalizability. Third, specimens collected and stored months or years prior to a clinical diagnosis (so called phase III specimens) are necessary to demonstrate the model's potential in the early detection of EOC in a general population or elevated risk setting. For the former, we have access to PLCO specimens; and for the latter, we plan to apply for specimens to the National Clinical Trials Network. Fourth, a logical extension of our work is to determine whether our current miRNA panel (or a new one) would be useful in predicting survival after EOC. A tissue-based MiROvaR signature involving 35 miRNAs for predicting EOC prognosis has recently been described (*Bagnoli et al., 2016*). Although several miRNAs appear in the tissue signature and our model (*Supplementary file 4B*), full concordance is unlikely since the tissue model was built to predict prognosis whereas our model was built to predict diagnosis. In addition, about two-thirds of the miRNAs in the tissue signature are not reliably detectable in circulation, which can be attributed to the fact that relatively few miRNAs circulate in serum (*Mestdagh et al., 2014*; *Dinh et al., 2016*). Our serum panel is reliant on a smaller number of miRNAs simply because the neural model prioritizes ones that provide novel information. If miRNAs are correlated (for example within the same chromosomal cluster), they will be invariant and knowing one will convey sufficient information about the rest for them to be excluded from model building. Finally, more is to be learned about the basic

biology of serum miRNA. Are they all coming from cancer cells or also other cells in the tumor micro-environment? (Likely, both are included in the signature). It is noteworthy that two of the miRNAs are members of the mir-200 family, confirming prior reports identifying these miRNAs as overexpressed in ovarian cancer (*Zuberi et al., 2015*; *Pecot et al., 2013*). Some of the miRNAs incorporated into the neural network algorithm have connections to other disease types. For example, miR-1246 has been identified in the serum of ovarian cancer, lung cancer, prostate cancer, and stroke patients (*Todeschini et al., 2017*; *Zhang et al., 2016*; *Alhasan et al., 2016*; *Li et al., 2015*). However, as noted in *Figure 5*, the network as a whole was specific to ovarian cancer, again emphasizing the importance of multimarker panels.

In conclusion, serum miRNA adds to the toolbox of options to diagnose ovarian cancer. We plan several future studies to characterize the miRNA neural network. Whether serum miRNA offers a lead time advantage over other putative biomarkers remains to be proven. We need to study the performance characteristics of the miRNA neural network in high risk and low risk populations. Finally, we are performing laboratory investigations to elucidate the biologic function of these miRNAs and to understand the kinetics of miRNA expression in ovarian cancer pathogenesis. With our improved understanding of miRNA analytic approaches, we can develop better models for this and other diseases.

## Materials and methods

### Reporting guidelines
Results have been reported according to the Transparent reporting of a multivariable prediction model for individual prognosis or diagnosis (TRIPOD) guidelines (Reporting Standards Document) (*Collins et al., 2015*). The checklist appears in the Supplement.

### Study subjects for model development
Our model was developed from two 'phase II' specimen sets (i.e. samples collected from women prior to surgery or chemotherapy) - Effects of Regional Analgesia on Serum microRNAs after Oncology Surgery (ERASMOS) and the Pelvic Mass Protocol (*Cramer et al., 2010*; *Elias et al., 2015*). To these, healthy controls were selected from subjects who participated in the New England Case-Control (NECC) study, a large epidemiologic study matching cases of ovarian cancer to geographically situated controls (*Rice et al., 2013*). These studies were approved by the Dana-Farber Cancer Institute Institutional Review Board Protocol 05–060 (NECC study), Brigham and Women's Hospital Institutional Review Board Protocol 2000-P-001678 (Pelvic Mass Protocol), and Dana-Farber/Harvard Cancer Center Institutional Review Board Protocol 12–532 (ERASMOS). All subjects were enrolled after signing informed consent, and samples were collected fresh in 13 × 75 mm BD Vacutainer Plus Plastic Serum tubes (BD Life Sciences, Franklin Lakes, NJ) with spray-coated silica. Samples were allowed to clot 1 hr at room temperature before processing, then spun down by centrifugation at 1300 x g x 10 min, aliquoted into 1.5 ml vials and stored at – 80 C. Samples from the other studies were thawed and aliquoted for the current study and then refrozen.

ERASMOS enrolled 60 patients from 03/2013 – 05/2015 from the Gynecologic Oncology service at DFCI and BWH. Patients were approached consecutively for enrollment. Eligible patients were scheduled to undergo exploratory laparotomy for a pelvic mass suspicious for invasive epithelial ovarian cancer. Serum blood samples were collected preoperatively and postoperatively for each patient and then stratified for analysis by anesthetic and analgesic exposure. The primary endpoint of the study is overall survival; study results have not been published to date as the final data are not mature.

The Pelvic Mass Protocol (PMP) enrolled women referred to the DFCI/BWH Gynecologic Oncology service over the period 1992 to 2013 (*Williams et al., 2014*). Of some 455 women with a pelvic mass enrolled, we selected a total of 120 samples from the following categories: serous cystadenoma (*Samuel and Carter, 2016*), serous borderline tumor (*Samuel and Carter, 2016*), Stage I/II invasive serous adenocarcinoma (*Häusler et al., 2010*), and Stage III/IV invasive serous adenocarcinoma (*Wang et al., 2016*), endometrioma (*Samuel and Carter, 2016*), Stage I/II invasive clear cell or endometrioid adenocarcinoma (*Häusler et al., 2010*), or Stage III/IV invasive clear cell or endometrioid adenocarcinoma (*Wang et al., 2016*). Overall, 37% of the subjects had benign disease, 12.6%

had borderline tumors, 10.1% had low grade carcinomas, and 40.4% had high grade carcinomas. One sample of serous cystadenoma was excluded as an outlier due to a recent cardiovascular event as evidenced by extreme elevation of myocardial ischemia-associated miRNAs. From the most recent phase (2004–2008) of the NECC study, we selected fifteen age and race matched healthy controls matched to the demographics of the EOC cases and benign disease controls from the PMP study. There was no overlap of subjects between the two studies. The samples sizes were based on a plan for a 2:1 ratio of early stage (Stage I/II) cancer cases to advanced stage (Stage III/IV) cases, a 1:1 ratio of invasive cancer cases: benign/borderline/control subjects, and for balanced numbers of healthy control: benign serous: benign endometrioid: borderline serous subjects. Borderline endo-metrioid or clear cell tumors were exceedingly rare and thus not included. For the qPCR model, we added 113 epithelial ovarian cancer cases and 113 healthy controls, matched for age and collection year. 20 failed quality control, leaving 206 additional samples to add to the 119 samples originally profiled from PMP and creating a 325 sample set for qPCR-based model building and cut-off calibration.

## Study subjects for external validation

Serum samples were collected from consecutive women undergoing surgical evaluation at the Medical University of Lodz, Poland, for a pelvic mass in association with an IRB-approved tumor collection protocol. All subjects were enrolled after signing informed consent, and samples were collected fresh in 13 × 75 mm BD Vacutainer Plus Plastic Serum tubes (BD Life Sciences, Franklin Lakes, NJ) with spray-coated silica. Samples were allowed to clot 1 hr at room temperature before processing, then spun down by centrifugation at 1300 x g x 10 min, aliquoted into 1.5 ml vials and stored at – 80 C. Samples were thawed only for the present study.

## Outcome

Samples were classified as either invasive cancer or benign/borderline/controls. Although borderline tumors are not strictly benign, they are clinically indolent and seldom fatal, thus we grouped them with benign lesions as our goal was to diagnose the tumors most contributing to mortality. For each patient, an estimated probability of >0.5 was classified as predicting invasive ovarian cancer.

## Next generation sequencing

For next generation sequencing (NGS), sample preparation, library construction, and miRNA sequencing were performed by Exiqon, Inc. (Vedbæk, Denmark). 500 μl of human serum from each sample were analyzed in duplicate. RNA from each serum sample was isolated using the miRCURY$^{TM}$ RNA isolation kit (Exiqon, Vedbæk, Denmark) per the manufacturer's protocol optimized for serum. The quality of the isolated RNA was checked using qPCR. Total RNA was converted into microRNA NGS libraries using the NEBNEXT library generation kit (New England Biolabs Inc., Ipswich, MA) per the manufacturer's instructions. Each individual RNA sample had adaptors ligated to its 3' and 5' ends and converted into cDNA. Then the cDNA was pre-amplified with specific primers containing sample-specific indices. After 18 cycles of pre-PCR the libraries were purified on QiaQuick columns and the insert efficiency evaluated by a Bioanalyzer 2100 instrument on a high sensitivity DNA chip (Agilent Inc., Lexington, MA). The microRNA cDNA libraries were size fractionated on a LabChip XT (PerkinElmer Waltham, MA) and a band representing adaptors and 15–40 bp insert excised using the manufacturer's instructions. Samples were then quantified using qPCR and concentration standards. Based on the quality of the inserts and the concentration measurements, the libraries were pooled in equimolar concentrations (all concentrations of libraries to be pooled were of the same concentration). The library pools were finally quantified again with qPCR and the optimal concentration of the library pool used to generate the clusters on the surface of a flowcell before sequencing using v3 sequencing methodology according to the manufacturer instructions (Illumina Inc., Dedham, MA). Samples were sequenced on the Illumina NextSeq 500 system (Illumina Inc., Dedham, MA) using a single-end read length of 50 nucleotides at an average of 10 million reads per sample. Sequence tags were mapped to miRbase 20 (http://www.mirbase.org/). After sequencing adapters were trimmed off as part of the base calling, trimming of adapters from the dataset revealed distinct peaks representing microRNA (~18–22 nt). Putative microRNAs not in standard miRBase or Rfam classification were identified based on the prediction algorithm miRPara and are included with the

sequencing data in the GEO file (*Wu et al., 2011*). Expression levels were quantified in tags per million (TPM) (unadjusted data and batch-adjusted data available in *Supplementary file 6*). TPM is a unit used to measure expression in NGS experiments. The number of reads for a particular miRNA is divided by the total number of mapped reads and multiplied by 1 million. Raw sequencing data are accessible as. fastq files through the Gene Expression Omnibus (GEO) database, www.ncbi.nlm.nih. gov/geo Accession GSE94533. The most stable miRNAs from the sequencing data were selected as normalizers using the NormFinder algorithm (*Andersen et al., 2004*).

### qPCR

miRNAs incorporated into the final neural network model were confirmed using qPCR with Exiqon (Vedbæk, Denmark) LNA-containing miRNA-specific probes. We selected nine potential reference miRNAs (hsa-miR-423–3 p, hsa-miR-103a-3p, hsa-miR-222–3 p, hsa-miR-221–3 p, hsa-miR-191–5 p, hsa-miR-181a-5p, hsa-miR-148b-3p, hsa-miR-146b-5p, and hsa-let-7c-5p) from the miRNA sequencing data using the NormFinder algorithm (*Andersen et al., 2004*). Both the 14 miRNAs from the test set and nine potential reference miRNAs were profiled using Exiqon's pick-and-mix array with LNA-containing miRNA-specific probes. Small RNA from each serum sample was isolated using the miRCURY RNA isolation kit (Exiqon, Vedbæk, Denmark) per the manufacturer's protocol optimized for serum. The quality of the isolated RNA was checked using qPCR. All miRNAs were polyadenylated and reverse transcribed into cDNA in a single reaction step. cDNA and ExiLENT SYBR Green master mix were transferred to qPCR panels pre-loaded with primers using a pipetting robot. Amplification was performed using a Roche Lightcycler 480 (Roche, Basel, Switzerland). Amplification quality was determined by generating melting curves. Raw Cq values and melting points, detected by the Lightcycler software, were exported. Assays with several melting points or with melting points deviating from assay specifications were flagged and removed from the dataset. Reactions with amplification efficiency below 1.6 were also removed. Assays giving Cq values within 5 Cq values of the negative control sample were also removed from the dataset.

Spike-in positive controls and no template negative controls were included. Minimum detection values for qPCR were established at 37 cycles; miRNAs with no amplification before that number of qPCR cycles were assumed to have their expression undetectable, and a quantification cycle (Cq) value of 37 was imputed as a substitute value. Raw, background filtered, and normalized data appear in the supplement (*Supplementary file 6*) in accordance with Minimum Information for Publication of Quantitative Real-Time PCR Experiments (MIQE) Guidelines (*Bustin et al., 2009*). Data were normalized to the average of the assays detected in all samples (n = 120 samples). The nine selected reference miRNAs were reevaluated after profiling for their stability across the arrays and the average Cq of the two best ones (miR-423–3 p and miR-103a-3p) was selected as the reference for dCq calculations of the 14-miRNA and the 7-miRNA diagnostic sets using the NormFinder method.

### Comparison of preoperative and postoperative samples

Individual miRNAs measurements from preoperative and postoperative serum samples from the ERASMOS study had been measured previously using multiplexed miRNA hydrogel probes (FirePlex, Abcam, Cambridge, MA) on a flow cytometer. Samples were profiled in duplicate, then replicates were merged. Fluorescence intensity values across all samples were normalized with Firefly Analysis Workbench (Abcam, Cambridge, MA) using the geNorm algorithm to identify appropriate normalizers (*Vandesompele et al., 2002*).

### Sample size estimation

We sought a testing set showing a superiority of 0.1 in the area under the receiver operating characteristic curve (AUC) against a value of 0.75 (assumed as a null hypothesis for a clinically useful biomarker) with a statistical power of 80% and a type 1 error probability <0.05 (*Hanley and McNeil, 1982*). For statistical power estimation purposes we assumed that the model predictions would be moderately correlated with CA-125 levels (r > 0.3). The calculation yielded a required testing set of 44 patients (22 with invasive cancer and 22 without invasive cancer). To train the classifiers, we assumed the training set would require 3-fold more patients (N = 132) bringing the total number of

required patient samples to 176 samples. We increased the sample size to 180 to account for potential clinical or technical outliers.

## Model development

### Variable selection

miRNAs were filtered for miRNAs present in at least 50% of both datasets at a detection threshold of 10 tags per million reads (tpm), leaving 192 miRNAs to test in our models. The data were batch-adjusted using ComBat to account for the different study populations (*Figure 9*; *Supplementary file 6*) (*Johnson et al., 2007*). Subject samples were then randomized into 'training' and 'testing' sets in an approximate 70:30 ratio.

As the dataset included more variables than cases, direct model development on the full dataset would have resulted in overfitted results. Hence, typically for such data mining problems we preselected the variables for classification model development using three methods – a significance filter (using a student's t-test – *Supplementary file 6*), a group-stratified fold change filter, and a correlation-based feature selection (CFS) (*Witten, 2016*). For the significance filter, we assumed miRNAs with p<0.05 and false discovery rate (FDR) < 0.05 for cancer vs benign/borderline/controls as significant. For the fold change filter, we selected miRNAs that showed fold changes < 0.8 or>1.2 for cancer vs benign/borderline/control comparisons convergent in both the PMP/NECC and ERASMOS datasets. Correlation-based Feature Subset Selection (CFS) is a wrapper feature selection method that evaluates the worth of a subset of attributes by considering the individual predictive ability of each feature along with the degree of redundancy between them. Subsets of features that were highly correlated with the class while having low intercorrelation were preferred in the process (*Hall, 1998*). Search of the space of attribute subsets was performed by greedy hillclimbing

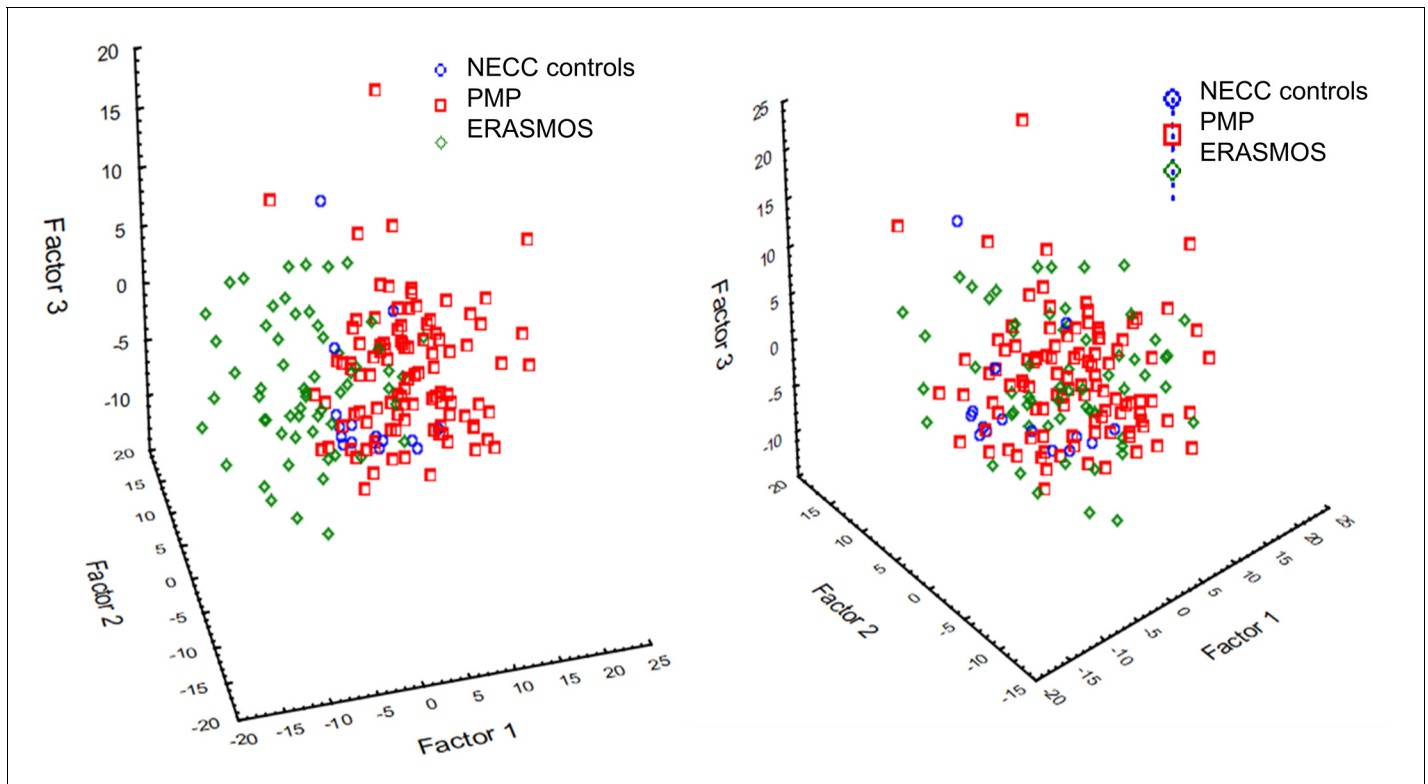

**Figure 9.** Principal component analysis identified a prominent batch effect among the study populations. (Left) Before batch effect removal. (Right) After batch effect removal using ComBat . ERASMOS – Effects of Regional Analgesia on Serum miRNA after Oncology Surgery Study. PMP – Pelvic Mass Protocol. NECC – New England Case Control study.
DOI: https://doi.org/10.7554/eLife.28932.019

augmented with a backtracking facility. This method of searching, called 'Best First', started with the empty set of attributes and searched the set forward.

## Classification models

All three sets of variables were analyzed using 11 different classification models for a total of 33 different algorithms. Six models (linear discriminant analysis, logistic regression, multivariate adaptive regression splines, naive Bayes, neural network, and support vector machine) were developed using STATISTICA Data Miner 12.5 (StatSoft, Tulsa, OK, USA). The remaining five models (functional tree, LAD tree, Bayesian network, elastic net regression, and random forest) were created using Weka 3.9.0 (University of Waikato, New Zealand). Detailed descriptions of the classification models appear below. Interestingly, relationships among individual miRNA species were non-linear, so these relationships would likely have been obscured as evidenced by a simple hierarchical clustering of the statistically significant miRNAs from univariate analysis (*Figure 10*).

All three sets of variables were analyzed using 11 different classification models for a total of 33 different algorithms. Six models (linear discriminant analysis, logistic regression, multivariate adaptive regression splines, naive Bayes, neural network, and support vector machine) were developed using STATISTICA Data Miner 12.5 (StatSoft, Tulsa, OK, USA). The remaining five models (functional tree, LAD tree, Bayesian network, elastic net regression, and random forest) were created using Weka 3.9.0 (University of Waikato, New Zealand). Interestingly, relationships among individual miRNA species were non-linear, so these relationships would likely have been obscured as evidenced by a simple hierarchical clustering of the statistically significant miRNAs from univariate analysis (*Figure 7*).

## Neural network

For the neural network, we built 5000 neural networks for each variable selection method (15000 networks in total) and retained the best one in terms of performance in properly assigning cases to classes in the test set. The networks were built in a semi-automated way. Their structure was of a multilayer perceptron with a number of neurons in the hidden layer iteratively optimized from (n variables)/3 to (n variables)*1.5 to avoid overfitting. Admissible linking functions between the neuron layers were linear, logistic, hyperbolic tangential, and exponential. Neuron weights were calculated using the BFGS (Broyden-Fletcher-Goldfarb-Shanno) algorithm and the network was trained in each

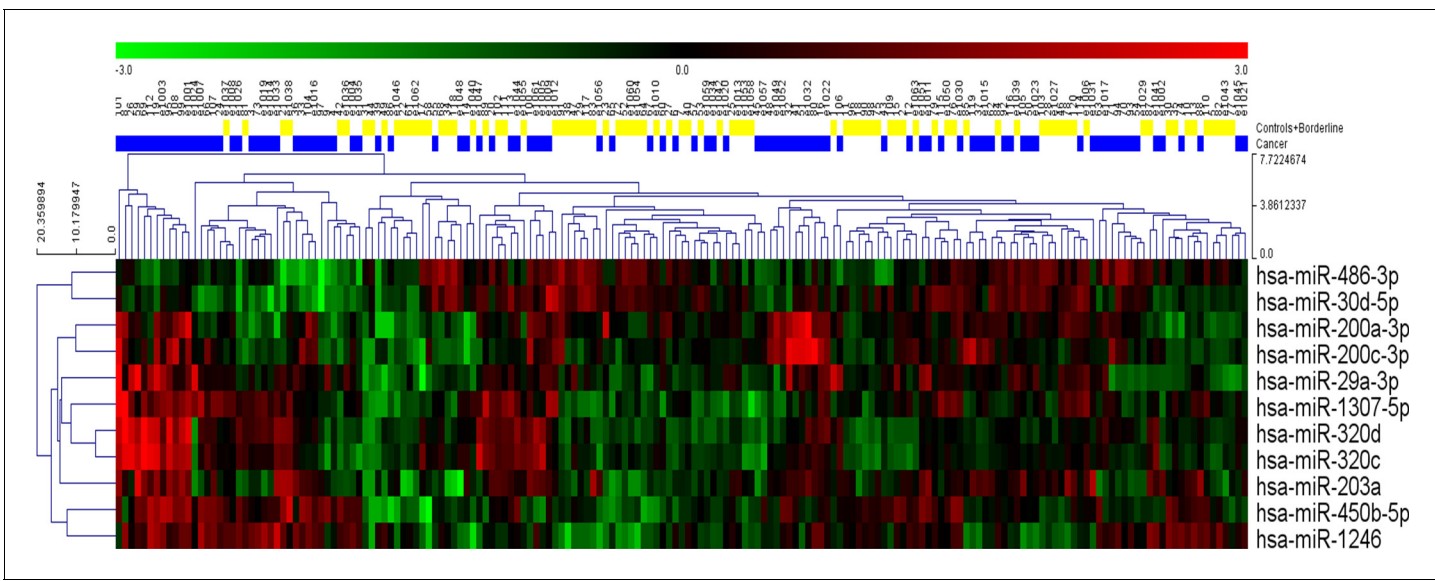

**Figure 10.** Hierarchical clustering of the eleven statistically significant miRNAs identified using univariate analysis. While most of the patients with cancer clustered together, considerable heterogeneity was evident, and no clear separation of the groups could be achieved using any single miRNA.
DOI: https://doi.org/10.7554/eLife.28932.020

epoch using an error back-propagation algorithm to optimize weights in each pass (*Broyden, 1970*; *Fletcher, 1970*; *Goldfarb, 1970*; *Shanno, 1970*; *Shanno and Kettler, 1970*).

## Linear discriminant analysis

The method creates a new set of spatial coordinates that allow for linear separation of the groups. The most discriminative features were extracted on the basis of their correlations and the model used a backward stepwise variable selection algorithm only retaining in the model variables that showed final F values > 5. This two-step filtering (variable selection after one of the three initial variable filtering algorithms) of the variables used in sample classification was aimed at the reduction of the number of miRNAs required for the model to work. Depending on the number of variables selected by the filters, the discriminatory function of the LDA was based on a reduced set of miRNAs that passed the F value threshold and were retained in the model. For the subset of miRNAs filtered by statistical significance, the model used three miRNAs: miR-30d-5p, miR-200c-3p and miR-320d. For CFS variable selection the model used three miRNAs: miR-320d, miR-200a-3p and miR-16-2-3p. The variable selection method based on stratified fold change used a yet another different set of miRNAs: miR-200c-3p, miR-320d and miR-150–5 p.

## Logistic regression

As above, the logistic regression model was built using a backward stepwise variable selection procedure, with variables showing p<0.15 being retained in the final model. The procedure allowed for second order interactions between the variables to detect potential subgroup-specific effects. A standard quasi-Newton estimation procedure was performed in model development. After exclusion of variables with p values > 0.15 in the multivariate model, the miRNAs remaining in the classifier were miR-30d-5p, miR-320d, miR-200c-3p, miR-1246, and an interaction of miR-200c-30p*miR-1246. A logistic regression model based on miRNAs selected by the CFS variable algorithm required only two miRNAs to work: miR-200c-3p and miR-320d. A logistic regression classifier built on the fold change filter-selected miRNAs used three miRNAs: miR-150–5 p, miR-320d, miR-1246, and an interaction between miR-200c-30p*miR-1246. Results of all three models were convergent and the crucial role of miR-200c/miR-320d was confirmed by all models. The logistic regression model was very similar in terms of performance to the neural net in the CFS-selected variable subset. This was a logical consequence of a strong variable filtering leaving too few input variables for the network to identify subtle patterns.

## Multivariate adaptive regression splines

An alternative approach to modeling of the classification function was the MARS model – a modification of a multivariate joint-point regression which estimates a number of basal function most appropriate for data from specific fragments of the multidimensional dataset. The method is used in complex function modeling of non-monotonous or non-linear associations. Within our analysis we used a MARS model that allowed for up to third degree interactions between the variables, allowing for up to 1.5*(n variables) basal function in each model and penalizing the introduction of additional basal functions by a factor of 2. Interactions between variables were tested for improvement of model performance up to the degree of three. During the model building procedure we iteratively removed variables absent in any of the basal functions until only miRNAs used in at least one basal function remained in the MARS model. Using 11 miRNAs filtered on the basis of significance we created a MARS model composed of 14 basal functions. All functions were transformation of five, single miRNAs: miR-30d-5p, miR-200c-3p, miR-450b-5p, miR-200a-3p, and miR-1307–3 p. The MARS model built on CFS-filtered variables consisted of 7 basal functions based on four miRNAs: miR-200c-3p, miR-320d, miR-16-2-3p, and miR-320b. The final MARS model built on 14 miRNAs filtered by the stratified fold change threshold was optimized at 10 basal functions based on 5 miRNAs: miR-200c-3p, miR-150–5 p, miR-200a-3p, miR-92–3 p, miR-203a, and miR-320c. All MARS models showed relatively poor performance hinting at issues with model overfitting and low specificity (for example, the ROC AUC for the significance-based and CFS variable selection inputs did not meet statistical significance).

## Elastic net regression

An elastic-net regularized generalized linear model is a linear regression using coordinate descent. In order to train this model we have used Java implementation of a component of the R package 'glmnet' in WEKA software. As we wanted to use a regression method for classification, class was binarized and one regression model was built for each class value (i.e. meta-scheme classification via regression). The alpha elastic-net mixing parameter was chosen to be 0.001 while the epsilon value for generating the lambda sequence was set to $10^{-4}$. Additionally, a covariance update method was used. This resulted in the following formula: *weka.classifiers.meta.ClassificationViaRegression -W weka.classifiers.functions.ElasticNet − -m2 y -alpha 0.001 -lambda_seq -thr 1.0E-7 -mxit 10000000 -numModels 100 -infolds 10 -eps 1.0E-4 -sparse n -stderr_rule n -addStats n*. Please note that reproduction of model induction may require installing additional packages from WEKA package manager.

Elastic net is a type of linear modeling. As so, application of classification via regression resulted in construction of 2 linear functions equations and as the class was binary – those equations had equally opposite coefficients. For example, classifier for class cancer in CFS-based dataset was based on the equation:

$$P(17) = -0.110\,hsa - miR - 16 - 2 - 3p + 0.050\,hsa - miR - 200a - 3p + 0.275\,hsa - miR - 200c - 3p$$
$$+ 0.043\,hsa - miR - 320b + 0.261\,hsa - miR - 320d - 0.031$$

Model files can be loaded in WEKA for further evaluation.

## Support vector machine

This classifier was built with a set of different entry parameters: kernel function types, function parameters, and hinge loss function. Admissible kernel functions were linear, polynomial (2$^{nd}$ and 3$^{rd}$ order) and radial basis function (gamma from 0.1 to 1 tested in 0.1 increments).All possible combinations were tested and the resulting best model was selected on the basis of classifier performance in the test set. All SVM models codes for significance, CFS and stratified fold change-based variable selection algorithms are available as pmml files. The models performed worse than simpler classification tools (logistic regression/linear discriminant analysis), possibly due to a small number of cases available for testing.

## Naïve bayes classifier

*A priori* class probabilities were estimated empirically on the basis of class frequencies in the dataset, normal distribution was assumed for all log-10 transformed miRNA expression values quantified as transcripts per million. The exact probability estimator of the naïve Bayes classifier showed similar performance on all three variable subsets, achieving accuracy comparable to that of the SVM model

## LAD tree

Multi-class alternating decision tree using the LogitBoost strategy (LAD Tree [http://www.cs.waikato.ac.nz/~bernhard/papers/ecml2002.pdf]). The number of boosting iterations to use, which determined the size of the trees, was set to be 10.

Formula: *weka.classifiers.trees.LADTree -B 10*. Please note that reproduction of model induction may require installing additional packages from WEKA package manager.

LADTree is a completely deterministic tree that allows decision making by counting respective probabilities on the pathway though the tree. Those trees and probabilities are available as buffer text files and WEKA model files. It is notable, that our configuration allowed the algorithm to consider a maximum of 10 miRNAs in the final schema.

## Functional tree

Functional trees are logistic classification decision trees that have logistic regression functions at the inner nodes or leaves. Training of models was performed again by WEKA software. As in default settings, the minimum number of instances at which a node is considered for splitting was 15, number of iterations for LogitBoost was also 15 and no weight trimming was applied.

Formula: *weka.classifiers.trees.FT -I 15 F 0 -M 15 -W 0.0*. Please note that reproduction of model induction may require installing additional packages from WEKA package manager.

All functional trees were models with one node. In order to infer how this model works, evaluation of values for linear combination function at each node for every class has to be done. For example, for cancer in the CFS-processed dataset the formula is:

$$F1 = -1.75 + [hsa-miR-16-2-3p] * -0.29 + [hsa-miR-200a-3p] * 0.08 + [hsa-miR-200c-3p] * 1.07 + [hsa-miR-320b] * -0.21 + [hsa-miR-320d] * 1.29$$

F1 = −1.75 + [hsa-miR-320d] * 1.29

As our classifiers are binary, the result for the second class (F2) should be an opposite number (F1 = -F2). In the next step the value of the following formula should be calculated and compared to threshold of the node:

$$\frac{e^{F1}}{e^{F1} + e^{F2}}$$

Model files can be loaded in WEKA for further evaluation.

## Bayesian network

A Bayes Network was trained using a K2 search algorithm, which is a hill climbing algorithm restricted by an order on the variables. The initial network used for structure learning was a Naive Bayes Network and there could be only one parent a node. Conditional probability tables of a Bayes network were driven directly from data once the structure has been learned (with alpha value equal to 0.5). Formula: *weka.classifiers.bayes.BayesNet -D -Q weka.classifiers.bayes.net.search.local.K2 – -P 1 -S BAYES -E weka.classifiers.bayes.net.estimate.SimpleEstimator – -A 0.5*. Please note that reproduction of model induction may require installing additional packages from WEKA package manager. Structures of networks as well as LogScores are available as buffer text files. Model files can be loaded in WEKA for further evaluation.

## Random forest

Random forest is a technique of random decision forests that considers K randomly chosen attributes at each node. K was calculated as integer of 1 plus binary logarithm of number of predictors. Minimum proportion of the variance needed at a node in order for splitting to be performed was set to 0.001. No backfitting was performed.

Formula: *weka.classifiers.trees.RandomForest -P 100 -I 100 -num-slots 1 -K 0 -M 1.0 -V 0.001 -S 1*. Please note that reproduction of model induction may require installing additional packages from WEKA package manager. Random forest is a form of bagging with 100 iterations and base learner. Model files can be loaded in WEKA for further evaluation.

## Basic statistical analysis

Differences in the distribution of histopathologic diagnoses, grade, and stage between the study populations were calculated using chi-square tests. Differences in false-positive and false-negative assignment were compared using Fisher's exact test. Differences in age and CA125 levels between the study populations were calculated using a Mann-Whitney U test. For all tests, a two-tailed p-value<0.05 was considered significant. For the ROC curves, cut-off values for prediction with the best diagnostic performance were established using the Youden index (sensitivity$_c$ + specificity$_c$ – 1) (*Youden, 1950*). Preoperative and postoperative serum samples from patients enrolled in ERASMOS were compared using a Wilcoxon matched pairs sign rank test.

## Code availability

Computer codes are available as raw pmml files in the supplement.

## Public dataset

The neural network approach was applied to an independent, publicly available published dataset by Keller, *et al.* GEO Accession GSE31568 (*Keller et al., 2011*) In that study, the authors collected blood samples from 454 individuals, including 15 women with ovarian cancer and 70 healthy controls. Further clinical annotation of the samples was not provided. The samples include a variety of other diagnoses (stomach cancer, sarcoidosis, prostate cancer, periodontitis, pancreatitis, pancreatic

cancer, multiple sclerosis, melanoma, lung cancer, chronic obstructive pulmonary disease, Wilms tumor, and acute myocardial infarction). Circulating miRNAs were quantified using a highly specific primer extension–based microarray that shows a very small degree of cross-hybridization (*Supplementary file 6*) (*Vorwerk et al., 2008*).

## Pathology samples

Paraffin blocks were selected from the surgical pathology files of the Brigham and Women's Hospital per BWH IRB Protocol #2016P002742. Hematoxylin and eosin sections of the cases were reviewed by a gynecologic pathologist (CC). The tissues had been routinely fixed in 10% neutral formalin and embedded in paraffin. Immunohistochemistry for TP53 and Ki-67 were performed using commercially available antibodies as previously described (*Perets et al., 2013*). Appropriate positive and negative (without primary antibodies) controls were used simultaneously for each antibody. In situ hybridization was performed using commercially available RNA probes from Exiqon (Vedbæk, Denmark) according to the manufacturer's instructions. All probe concentrations were 1 nM. A probe for the small nuclear RNA U6 served as a positive control while a non-targeting scramble RNA probe served as negative control.

## Acknowledgement

The authors wish to acknowledge funding support from the Robert and Deborah First Family Fund (KME, RSB, DC), NICHD K12HD13015 (KME), the Ruth N White Research Fellowship in Gynecologic Oncology (KME, SJF, RSB), the Saltonstall Research Fund (KME, SJF, RSB), Potter Research Fund (KME, SJF, RSB), the Sperling Family Fund Fellowship (KME, SJF, RSB), the Bach Underwood Fund (KME, SJF, RSB), First TEAM grant of the Foundation for Polish Science and the Smart Growth Operational Programme of the European Union (WF), NIH P50CA105009 (DWC, AFV), Department of Defense grants OC093426 (PAK) and OC140632 (PAK), the William M Wood Foundation (DC) and the Honorable Tina Brozman Foundation (KME, DC).

## Additional information

### Funding

| Funder | Grant reference number | Author |
|---|---|---|
| Robert and Deborah First Family Fund | | Kevin M Elias<br>Ross S Berkowitz<br>Dipanjan Chowdhury |
| National Institutes of Health | K12HD13015 | Kevin M Elias |
| Ruth N White Research Fellowship in Gynecologic Oncology | | Kevin M Elias<br>Stephen J Fiascone |
| Saltonstall Research Fund | | Kevin M Elias<br>Stephen J Fiascone<br>Ross S Berkowitz |
| Ian Potter Foundation | | Kevin M Elias<br>Stephen J Fiascone<br>Ross S Berkowitz |
| Sperling Family Fund Fellowship | | Kevin M Elias<br>Stephen J Fiascone<br>Ross S Berkowitz |
| Bach Underwood Fund | | Kevin M Elias<br>Stephen J Fiascone<br>Ross S Berkowitz |
| Honorable Tina Brozman Foundation | | Kevin M Elias<br>Dipanjan Chowdhury |
| Fundacja na rzecz Nauki Polskiej | First TEAM grant | Wojciech Fendler |

| Smart Growth Operational Programme of the European Union | First TEAM grant | Wojciech Fendler |
| --- | --- | --- |
| Uniwersytet Medyczny w Lodzi | 502-03/1/-090-03/502-14-338 | Wojciech Fendler |
| National Institutes of Health | P50CA105009 | Allison F Vitonis<br>Daniel W Cramer |
| U.S. Department of Defense | OC093426 | Panagiotis Konstantinopoulos |
| U.S. Department of Defense | OC140632 | Panagiotis Konstantinopoulos |

The funders had no role in study design, data collection and interpretation, or the decision to submit the work for publication.

### Author contributions
Kevin M Elias, Conceptualization, Resources, Data curation, Formal analysis, Supervision, Funding acquisition, Investigation, Visualization, Methodology, Writing—original draft, Project administration, Writing—review and editing; Wojciech Fendler, Data curation, Software, Formal analysis, Methodology, Writing—review and editing; Konrad Stawiski, Software, Formal analysis; Stephen J Fiascone, Investigation; Allison F Vitonis, Gyorgy Frendl, Magdalena Kedzierska, Resources; Ross S Berkowitz, Resources, Writing—review and editing; Panagiotis Konstantinopoulos, Writing—review and editing; Christopher P Crum, Resources, Formal analysis; Daniel W Cramer, Resources, Supervision, Methodology, Writing—review and editing; Dipanjan Chowdhury, Conceptualization, Resources, Data curation, Formal analysis, Supervision, Funding acquisition, Validation, Investigation, Methodology, Writing—original draft, Project administration, Writing—review and editing

### Author ORCIDs
Kevin M Elias [iD] http://orcid.org/0000-0003-1502-5553
Wojciech Fendler [iD] http://orcid.org/0000-0002-5083-9168
Konrad Stawiski [iD] http://orcid.org/0000-0002-6550-3384
Dipanjan Chowdhury [iD] http://orcid.org/0000-0001-5645-3752

### Ethics
Human subjects: All samples were collected after signed informed consent according to locally-approved Institutional Review Board protocols. These studies were approved by the Dana-Farber Cancer Institute Institutional Review Board Protocol 05-060 (NECC study), Brigham and Women's Hospital Institutional Review Board Protocol 2000-P-001678 (Pelvic Mass Protocol), and Dana-Farber/Harvard Cancer Center Institutional Review Board Protocol 12-532 (ERASMOS).

### Decision letter and Author response
Decision letter https://doi.org/10.7554/eLife.28932.037
Author response https://doi.org/10.7554/eLife.28932.038

## Additional files

### Supplementary files
• Source code 1. miRNA-seq neural network source code.
DOI: https://doi.org/10.7554/eLife.28932.021

• Source code 2. qPCR 14-miRNA neural network source code.
DOI: https://doi.org/10.7554/eLife.28932.022

• Source code 3. qPCR 7-miRNA neural network source code.
DOI: https://doi.org/10.7554/eLife.28932.023

• Source code 4. neural network applied to the Keller, et al dataset.
DOI: https://doi.org/10.7554/eLife.28932.024

• Supplementary file 1. Performance of the various prediction models on the unadjusted datasets. (A) Area under the ROC curve analyses for the various testing methods depending on the variable selection protocol using data without batch adjustment. Like the batch-adjusted data, the neural network using the fold change variable outperformed the other methods in terms of classifier accuracy and did not overfit the predictions to the training set. (B) Individual sample predictions of the tested classification models built on the unadjusted fold change-based variable selection miRNA subset.

DOI: https://doi.org/10.7554/eLife.28932.025

• Supplementary file 2. Post-hoc secondary analyses of the neural network. (A) Misclassification matrices for the neural network and CA125 predictions with detailed histopathological data. (B) Misclassification matrices for the neural network stratified by age. (C) miRNA expression by tumor histology and stage.

DOI: https://doi.org/10.7554/eLife.28932.026

• Supplementary file 3. Characteristics of CA125 expression in the study populations. (A) Serum CA125 measurements among cancer and non-cancer cases in the two study populations. (B) Relationship between CA125 and miRNAs in the neural network.

DOI: https://doi.org/10.7554/eLife.28932.027

• Supplementary file 4. Comparison of the neural network to existing datasets. (A) Mapping of the 14-miRNA dataset from the miRNA-sequencing study onto the GSE31568 dataset published by Keller, et al. (B) Comparison of the neural network (NN) classifier with the tissue-based MiROvaR signature by Bagnoli et al.

DOI: https://doi.org/10.7554/eLife.28932.028

• Supplementary file 5. Univariate comparison of miRNA average expression values between patients with cancer and patients in the benign/borderline/control group.

DOI: https://doi.org/10.7554/eLife.28932.029

• Supplementary file 6. Supplementary datasets. Supplementary Dataset (1) TPM data from miRNA sequencing. Supplementary Dataset (2) Batch adjusted, log10-transformed miRNA expression data, filtered for miRNA detection levels in both cohorts. Supplementary Dataset (3) qPCR Validation of neural network. Supplementary Dataset (4) Raw qPCR data. Supplementary Dataset (5) Background Filtered qPCR data. Supplementary Dataset (6) Normalized qPCR data. Supplementary Dataset (7) Normalized expression data from the Keller et al. dataset. Supplementary Dataset (8) Normalized expression data of preoperative and postoperative miRNA expression.

DOI: https://doi.org/10.7554/eLife.28932.030

• Transparent reporting form

DOI: https://doi.org/10.7554/eLife.28932.031

• Reporting standard 1

DOI: https://doi.org/10.7554/eLife.28932.032

## Major datasets

The following dataset was generated:

| Author(s) | Year | Dataset title | Dataset URL | Database, license, and accessibility information |
|-----------|------|---------------|-------------|--------------------------------------------------|
| Elias KM, et al | 2017 | Serum microRNA sequencing for diagnosis of invasive ovarian cancer | https://www.ncbi.nlm.nih.gov/geo/query/acc.cgi?acc=GSE94533 | Publicly available at the NCBI Gene Expression Omnibus (accession no: GSE94533) |

The following previously published dataset was used:

| Author(s) | Year | Dataset title | Dataset URL | Database, license, and accessibility information |
|-----------|------|---------------|-------------|--------------------------------------------------|
| Keller A, Leidinger P, Bauer A, Elsharawy A, Haas J, Backes C | 2011 | The human Whole miRNOme project version 1 | https://www.ncbi.nlm.nih.gov/geo/query/acc.cgi?acc=GSE31568 | Publicly available at the NCBI Gene Expression Omnibus (accession no: GSE31568) |

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
