## [Decision Letter]

[Editors’ note: this article was originally rejected after discussions between the reviewers, but the authors were invited to resubmit after an appeal against the decision.]

Thank you for submitting your work entitled "Diagnostic potential for a serum miRNA neural network for detection of ovarian cancer" for consideration by *eLife*. Your article has been reviewed by two peer reviewers, and the evaluation has been overseen by a Reviewing Editor and a Senior Editor. The reviewers have opted to remain anonymous.

Our decision has been reached after consultation between the reviewers. Based on these discussions and the individual reviews below, we regret to inform you that your work will not be considered further for publication in *eLife*.

Although all reviewers acknowledge the strengths of the topic of predicting early stage disease and the analysis, the consensus is that the work lacks sufficient novelty for the general audience of *eLife*. Specifically, there are a couple of prior publications on miR detection in serum of ovarian cancer patients, and we are not persuaded that the machine learning algorithms used here represent the level of innovation we require. Second, the biological underpinnings of the predictive miR signature are not addressed, nor is the source of the miRs (are they from tumor cells?). Third, in the absence of a biological hypothesis, we would expect further work to eliminate many of the caveats raised about clinical utility, tumor samples with various histologies, further validation, etc.

*Reviewer #1:*

The authors present a paradigm for serum detection of ovarian cancer. They employ a neural network to discriminate between cancer and noncancer specimens and demonstrate specificity with published data. The paper has many strengths including a range of ovarian cancer specimens of various histologic subtypes and stages. There is very nice demonstration of training, test and external validation. Enthusiasm is somewhat weakened, due to:

1) Multiple statistical algorithms are tested leading to some concern for overfitting during creation of the original model.

2) There is little to no detail about the histology and stage of the external validation specimens and the small sample size of only 15 ovarian cases for external validation remains hidden within the methods. It would be good to know the histology and stage and more details about the collection methods for the external validation samples.

3) It is not clear how the samples were collected in Keller et al. and whether they were all from newly diagnosed patients prior to treatment.

Overall, this seems like a very promising signature, but past experience would suggest some additional externally validated samples should be included.

*Reviewer #2:*

Elias and colleagues have evaluated the utility of circulating miRNA as a potential tool to assist in the diagnosis of ovarian cancer. They identify miRNAs and an algorithm that distinguishes individuals without ovarian cancer from patients with cancer.

There are many publications describing miRNAs as a biomarker in ovarian cancer. Although the authors mention some of these previous studies examining circulating miRNAs in the Introduction, it is unclear whether the miRNAs utilised in the neural network algorithm overlap with those described in previous studies. It would strengthen the manuscript to compare the findings to previous work more explicitly.

The manuscript could also be strengthened by examining the potential of the algorithm and miRNAs to predict prognosis, as a biomarker that can be utilised for diagnosis and prognosis is very desirable.

[Editors’ note: what now follows is the decision letter after the authors submitted for further consideration.]

Thank you for choosing to send your work entitled "Diagnostic potential for a serum miRNA neural network for detection of ovarian cancer" for consideration at *eLife*. Your letter of appeal has been considered by a Senior Editor and a Reviewing Editor, and we are prepared to consider a revised submission with no guarantees of acceptance.

We are particularly interested in the inclusion of data regarding the biological underpinnings of the miR signature and additional external validation on another cohort of patients. It will also be important to emphasize the novelty relative to prior published analyses of serum miRs in ovarian cancer. As a dataset but without a biological rationale for the set of enriched miRNAS, this work appears to be more appropriately considered as a Tools and Resources paper, rather than as a Research Article.

---

## [Author Response]

[Editors’ note: what now follows is the decision letter after the authors submitted for further consideration.]

Although all reviewers acknowledge the strengths of the topic of predicting early stage disease and the analysis, the consensus is that the work lacks sufficient novelty for the general audience of eLife. Specifically, there are a couple of prior publications on miR detection in serum of ovarian cancer patients, and we are not persuaded that the machine learning algorithms used here represent the level of innovation we require.

While other publications have discussed the potential for miRNA detection in serum, these studies have been inadequate due primarily to the selection of the platforms and the size of the cohorts.

First, biases inherent to focused miRNA expression profiling platforms significantly influence results. A study published in Nature Methods (Nat Methods. 2014 Aug;11(8):809-15) systematically compared 12 different miRNA expression platforms. There were significant differences in results from the identical samples profiled with different platforms. Specifically, for serum miRNAs they state, "we evaluated rates of miRNA detection in the serum RNA samples by counting those miRNAs detected in at least two of four replicates. Detection rates in serum RNA were much more variable between platforms, with up to a 12-fold difference between the highest and lowest number of detected miRNAs.” For this reason, our publication is *the first to utilize small RNA sequencing* for its discovery approach, which is focused sequencing with significant depth to detect miRNAs in serum from 179 patients. Our approach captures results for all miRNAs without the biases introduced by the number of pre-selected probes or primers on an array plus the added sensitivity of sequencing all small RNAs.

Second, a challenge in studying miRNAs is the high degree of similarity between the sequences. Some microRNA family members vary by a single nucleotide. For example, sequence differences between the let-7 family members vary from one to four nucleotides while miR-302 family members have a two or three nucleotide difference. They address this issue in the Nature Methods study as well, stating, "A cross-reactivity heatmap displaying signal intensity for mismatched miRNA combinations relative to signal intensity of the perfect match demonstrates major differences between platforms and between both miRNA families. One platform (miRCury (EX); Exiqon) showed absolute specificity for both miRNA families. Whereas most platforms showed little or no cross-reactivity between miR-302 family members, cross-reactivity between let-7 family members was markedly higher and predominantly occurred between members differing in only one nucleotide." We used the Exiqon platform for all our validation studies as they used the Locked-Nucleic Acid based method. The melting temperature difference because of a single nucleotide change can only be amplified faithfully using a LNA primer/probe.

Third, prior studies on serum miRNAs have failed to provide sufficiently powered discovery and validation sets. These studies have provided either no discussion of power, no validation set, or neither of the two. Our work stands apart by the 3:1 randomization of samples into samples for model building and validation. To this, we have increased the power even further by adding another 220 samples to the qPCR model validation set and a third independent cohort of 50 patients from Poland into another external validation set.

Second, the biological underpinnings of the predictive miR signature are not addressed, nor is the source of the miRs (are they from tumor cells?).

A biologically plausible and useful tumor biomarker should have two qualities: 1) expression should change if the tumor is removed and 2) the biomarker should appear in the cells of interest across all stages of the disease.

First, among the study samples included in our work, we had preoperative and postoperative blood samples for one cohort, the patients enrolled in the ERASMOS study. These were drawn 72 hours apart. Figure 7 of the revised version illustrates how expression of several of the key miRNAs from the neural network decreased rapidly following optimal cytoreduction surgery, suggesting that the miRNAs were coming from tumor cells.

Second, the cell of origin for most epithelial ovarian cancers is the secretory epithelia of the distal fallopian tube. In Figure 8 of the revised version, we show for the first time that miRNAs from the neural network are expressed in serous tubal intraepithelial carcinomas and high grade Stage I serous and endometrioid ovarian cancers. Abnormal miRNAs are co-localized with mutations in tp53 and areas of high proliferative index, as marked by Ki-67 expression, which are the defining biologic markers of early high grade epithelial ovarian cancer lesions. The early acquisition of miRNA changes in both pre-invasive lesions and pre-metastatic cancers underscores the biologic utility of our miRNA neural network to provide a useful diagnostic test for ovarian cancer at a clinically meaningful time point in the disease.

Third, in the absence of a biological hypothesis, we would expect further work to eliminate many of the caveats raised about clinical utility, tumor samples with various histologies, further validation, etc.

In the revised manuscript, we have added a new external validation set from Poland. Among these study subjects, the neural network correctly identified 100% of Stage I and II ovarian cancers across histologic subtypes. It is plausible that due to limited sampling for rare histologies such as mucinous type or transitional cell type the application of the neural network will be limited; nonetheless, for the three most common histologies – serous, endometrioid, and clear cell – the neural network has high accuracy for discriminating cases of ovarian cancer.

It will also be important to emphasize the novelty relative to prior published analyses of serum miRs in ovarian cancer.

A more extensive description of how our study is distinct from prior studies has been described in the Discussion section of the revised manuscript.

Reviewer #1:The authors present a paradigm for serum detection of ovarian cancer. They employ a neural network to discriminate between cancer and noncancer specimens and demonstrate specificity with published data. The paper has many strengths including a range of ovarian cancer specimens of various histologic subtypes and stages. There is very nice demonstration of training, test and external validation. Enthusiasm is somewhat weakened, due to:1) Multiple statistical algorithms are tested leading to some concern for overfitting during creation of the original model.

Our goal in showing several statistical approaches was to demonstrate the consistency among the models, rather than to emphasize the superiority of one model. Ultimately, the neural network method was chosen based on a slightly higher performance characteristic, but as noted by similar AUC values, other approaches would be expected to give similar results. Moreover, overfitting was minimized by running the model on both adjusted and unadjusted datasets. ROC curves for both are shown to indicate that the results were similar.

2) There is little to no detail about the histology and stage of the external validation specimens and the small sample size of only 15 ovarian cases for external validation remains hidden within the methods. It would be good to know the histology and stage and more details about the collection methods for the external validation samples.

This is a published dataset from another group. We do not have more annotation than what those authors provided. Our goal was to use this dataset to highlight the specificity of our neural network for ovarian cancer by looking at the performance of the neural network against 12 competing diagnoses, not to indicate that the signature was useful for identifying 15 cases of ovarian cancer. This point is emphasized in the revised version of the paper. We do provide annotated data on the third independent cohort of 51 patients from Poland, however, which appears as a new external validation set in the revised manuscript.

3) It is not clear how the samples were collected in Keller et al. and whether they were all from newly diagnosed patients prior to treatment.

Again, we have limited annotation data for Keller et al., but as noted, this does not influence the specificity question. However, among the new external validation cohort, all samples were from newly diagnosed patients prior to treatment.

Reviewer #2:Elias and colleagues have evaluated the utility of circulating miRNA as a potential tool to assist in the diagnosis of ovarian cancer. They identify miRNAs and an algorithm that distinguishes individuals without ovarian cancer from patients with cancer.There are many publications describing miRNAs as a biomarker in ovarian cancer. Although the authors mention some of these previous studies examining circulating miRNAs in the Introduction, it is unclear whether the miRNAs utilised in the neural network algorithm overlap with those described in previous studies. It would strengthen the manuscript to compare the findings to previous work more explicitly.

As noted in the comments to the editor, while several limited studies have been performed using miRNA profiling by array or qPCR, ours is the first study to use the higher fidelity approach of small RNA sequencing. Also, many of the other studies do not specify whether the -3p or -5p miRNA is being used and none of these prior works consider analyzing several miRNAs species in a multiplexed approach. Thus, a direct comparison is difficult. We do note that several of the key miRNAs in the neural network, notably the mir-200a and mir-200c species, have been linked to ovarian cancer generally, and in the supplemental data, indicate where overlap is seen between the serum neural network and tissue-based profiling efforts.

The manuscript could also be strengthened by examining the potential of the algorithm and miRNAs to predict prognosis, as a biomarker that can be utilised for diagnosis and prognosis is very desirable.

We provide the overlap between our set and the miROvAr dataset, which has been linked to prognosis, in the supplemental information (Supplementary file 4). The strongest predictors of prognosis in ovarian cancer are stage, residual disease after cytoreductive surgery, and response to platinum-based chemotherapy. In this study, residual disease and chemotherapy data were not available for patients in all the cohorts, so we cannot compare the miRNA signature to those variables. Moreover, many of our study subjects completed their therapy at outside centers, so the heterogeneity in the treatment data would be considerable, minimizing the utility of a prognostic analysis.